# Polymeric 3D-Printed Microneedle Arrays for Non-Transdermal Drug Delivery and Diagnostics

**DOI:** 10.3390/polym17141982

**Published:** 2025-07-18

**Authors:** Mahmood Razzaghi

**Affiliations:** Department of Mechanical Engineering, University of Victoria, Victoria, BC V8P 5C2, Canada; mahmoodrazzaghi@uvic.ca

**Keywords:** microneedle arrays, 3D printing, non-transdermal applications, drug delivery, diagnostics

## Abstract

Microneedle arrays (MNAs) are becoming increasingly popular due to their ease of use and effectiveness in drug delivery and diagnostic applications. Improvements in three-dimensional (3D) printing techniques have made it possible to fabricate MNAs with high precision, intricate designs, and customizable properties, expanding their potential in medical applications. While most studies have focused on transdermal applications, non-transdermal uses remain relatively underexplored. This review summarizes recent developments in 3D-printed MNAs intended for non-transdermal drug delivery and diagnostic purposes. It includes a literature review of studies published in the past ten years, organized by the target delivery site—such as the brain and central nervous system (CNS), oral cavity, eyes, gastrointestinal (GI) tract, and cardiovascular and reproductive systems, among other emerging areas. The findings show that 3D-printed MNAs are more adaptable than skin-based delivery, opening up exciting new possibilities for use in a variety of organs and systems. To guarantee the effective incorporation of polymeric non-transdermal MNAs into clinical practice, additional research is necessary to address current issues with materials, manufacturing processes, and regulatory approval.

## 1. Introduction

Microneedle arrays (MNAs) are medical devices composed of micron-sized needles that enable the delivery of therapeutic agents and are used in diagnostic applications [1,2,3]. They were initially developed for transdermal use, allowing them to bypass the stratum corneum while avoiding contact with deeper pain receptors. This makes it easy to give medications or vaccinations through the skin. This method has several benefits over regular hypodermic needles. These benefits include the ability to self-administer, better patient adherence, and a lower risk of infection [4,5]. Besides delivering medications, MNAs have been used to collect biofluids for diagnostic purposes, like interstitial fluid (ISF) [6,7,8,9,10].

Polymeric MNAs have received a lot of attention for their compatibility with the body and flexibility. They can be made to dissolve or break down in water, allowing them to safely dissolve or decompose inside the body. This ability removes sharp waste and helps control how drugs are released [11]. For instance, in an osteoporosis study, dissolvable polymeric MNAs showed better delivery of a monoclonal antibody compared to traditional subcutaneous injection. This led to improved therapeutic results [12]. This shows that polymeric MNAs may match or even surpass the effectiveness of standard injections in certain treatment situations.

The early stages of MNA research mainly focused on delivering drugs through the skin, including vaccine patches and insulin administration [13]. More recent developments have expanded this technology into uses beyond just skin applications [11]. In this case, “non-transdermal” means using MNAs to reach targets deeper in the body, such as in specific organs or tissues. For instance, there are MNA-based systems designed for drug delivery or diagnostic purposes in the brain, central nervous system (CNS), oral cavity, eyes, gastrointestinal (GI) tract, cardiovascular system, and reproductive organs. These new applications aim to take advantage of the less invasive nature of MNAs, allowing access to areas that usually require larger needles or surgeries [11]. Recent reviews highlight the increasing use of these methods, showing that MNA technology is quickly moving beyond skin-related applications to a broader range of biomedical uses [11,14].

At the same time, notable progress has been made in MNA fabrication methods. Although techniques such as micromilling, lithography, and injection molding are effective, they typically require intricate, multi-step procedures that are expensive and hinder large-scale production [15,16,17,18]. To address these limitations, this review emphasizes three-dimensional (3D) printing as a transformative approach for fabricating MNAs. By offering a flexible, rapid, and cost-efficient method, 3D printing has significantly advanced the production of MNAs [19,20]. Three-dimensional printing encompasses a range of techniques that build objects layer by layer, enabling precise and reproducible fabrication of MNAs with high accuracy [21]. Particularly effective 3D printing techniques for producing complex polymeric MNA structures layer by layer include digital light processing (DLP), stereolithography (SLA), and two-photon polymerization (2PP). The size and shape of needles can be precisely controlled thanks to these methods [18,19,22,23]. Because factors like needle height and tip sharpness have a significant influence on patient comfort and performance, MNAs need to be fabricated with high precision [15,16].

A key advantage of 3D printing technology is that it allows us to create truly personalized designs for individual patients. This means we can develop MNAs that are specifically designed to address unique patient requirements and treatment goals [24,25]. Furthermore, the advancement of low-cost, high-resolution 3D printers has significantly reduced the financial and technical barriers, facilitating the broader adoption of MNA technology in both research and clinical settings [9,26]. Three-dimensional printing facilitates the development of multifunctional MNAs by integrating internal microchannels or reservoirs, enabling precise drug release at target sites and supporting theranostic applications [9,27,28]. The result has been a wave of innovations in polymeric 3D-printed MNAs, exploring new shapes (e.g., lattice or biomimetic designs for higher drug loading), new polymer materials (including composite and stimuli-responsive resins), and integrated functionalities like sensing elements [29].

This review provides an overview of polymeric 3D-printed MNAs used for non-transdermal drug delivery and diagnostics. It focuses on their application in the brain/CNS, oral cavity, eye, GI tract, cardiovascular system, reproductive organs, and some other organs and tissues, highlighting important recent studies. We start by summarizing different 3D printing methods used to create polymeric MNAs and compare their capabilities. In the next section on application areas, we look at each use case in detail. We also discuss current challenges, such as biocompatibility and scalability, and suggest future research directions. By emphasizing polymer-based MNAs, the review points out their unique benefits, including biodegradability and flexibility. Ultimately, this review explores how polymeric 3D-printed MNAs are expanding the possibilities of MNA technology beyond the skin and shaping the future of drug delivery and diagnostic systems.

## 2. Fabrication of Polymeric 3D-Printed MNAs

### 2.1. Overview of 3D Printing Techniques

Three-dimensional printing has emerged as a revolutionary method for MNA fabrication, providing high precision and customization through layer-by-layer construction directed by computer-aided design (CAD) models. For polymeric MNA fabrication, several 3D printing techniques have been developed, each offering unique advantages [30,31]. Among these, the most widely used methods are fused deposition modeling (FDM), 2PP, DLP, and SLA. The main 3D printing techniques for creating polymeric MNAs are reviewed in this section.

SLA is one of the most commonly used 3D printing methods for MNA fabrication. It uses an ultraviolet (UV) laser to selectively solidify liquid photopolymers, producing highly detailed structures with smooth surface finishes [32]. SLA is especially effective for producing complex MNA designs. Its adaptability has made it widely applicable in biomedical fields, particularly in crafting precise drug delivery systems and diagnostic devices where accuracy and surface smoothness are essential [19,30]. SLA has been widely utilized in the fabrication of 3D-printed MNAs [16,32,33,34,35,36,37,38,39,40,41,42]. This technique has facilitated the fabrication of diverse MNA types—such as hollow, dissolvable, and solid MNAs—primarily for transdermal drug delivery applications [21,39,40,41]. Additionally, SLA has been used to create master molds for dissolvable MNAs intended for ocular drug delivery [37], as well as to produce coated MNAs for intradermal insulin administration [38]. SLA 3D-printed MNAs have been used for MNAs that facilitate blood-free detection of biomarkers, such as C-reactive protein and procalcitonin in ISF [16].

In a study, Kadian et al. [43] presented the development of a projection micro-stereolithography (PmSL), 3D-printed conducting MNA-based electrochemical point-of-care device for minimally invasive detection of chlorpromazine, a widely used antipsychotic drug. They used PmSL, which is a type of advanced SLA technology, to fabricate high-resolution constructs, like MNAs, followed by inkjet printing of carbon and silver inks to construct a three-electrode system directly on the microneedle (MN) tips. To increase sensitivity, carbon dots were added to the working electrode. Even when interferents were present, the electrochemical analysis showed high selectivity toward chlorpromazine, a low detection limit, and excellent linearity. Using phantom gel and artificial interstitial fluid, the system was successfully tested in skin-mimicking models, demonstrating 86% signal retention after 20 days and promising real-time performance. The potential for wearable, scalable, and reasonably priced biosensors for drug monitoring is demonstrated by this proof-of-concept. Their developed MNA system is shown in Figure 1 [43].

Another 3D printing method is DLP, which is also used for polymeric MNA fabrication. DLP has some similarities with SLA, but it cures whole layers of photopolymer resin simultaneously with the aid of a digital light projector [44]. While SLA uses a laser to cure resin point by point, DLP exposes and cures a layer of resin all at once using a projector. This technique speeds up the printing by a rapid layerwise fabrication process [45,46]. The DLP method is commonly used for 3D printing of MNAs [9,10,18,42,47,48,49,50,51]. This technology has a relatively high resolution, often down to the micron [52]. For instance, DLP-printed MNAs made of polyethylene glycol diacrylate (PEGDA) have been developed for on-demand drug delivery and the multiplex detection of biomarkers, including pH, glucose, and lactate levels in skin ISF [9]. Additionally, in vivo testing on mice showed that continuous glucose monitoring in ISF can be successfully achieved using solid MNAs produced with biocompatible, light-sensitive resins [49]. Moreover, DLP has facilitated the development of drug delivery systems employing MNAs made from biocompatible resins, enabling increased permeability of active compounds with molecular weights in the range of 600 to 4000 Da in buccal tissue [18].

Liquid crystal display (LCD) technology is another method employed for 3D printing of polymeric MNAs [41,53,54,55]. Similar to SLA and DLP, LCD is a vat polymerization technique that utilizes photopolymer resins to achieve the high accuracy required for fabricating intricate microstructures such as MNAs. What sets LCD apart is its ability to deliver satisfactory resolutions at significantly lower costs, making it an ideal choice for large-scale production of complicated objects without compromising precision or affordability [56]. Similar to DLP, the LCD method generally prints at a faster rate than SLA, as it cures an entire layer simultaneously rather than tracing each point individually. However, LCD 3D printing fundamentally differs from DLP in its approach to light projection. While DLP uses a digital micromirror device to project the entire image of a layer, LCD employs a liquid crystal display panel that shines light through its pixels to cure the resin layer by layer. This method minimizes pixel distortion, ensuring consistent quality, but it typically requires slightly more time for curing compared to DLP due to its reliance on pixel-by-pixel illumination [57].

Static optical projection lithography (SOPL) is another 3D printing method that has been used for the fabrication of polymeric MNAs. This technique uses a fixed projection of digital light to selectively trigger polymerization in monomer solutions based on how the light intensity is distributed across the surface. Unlike traditional methods, it does not rely on any mechanical movement, allowing for extremely fast production of MNAs. Additionally, by modifying the projected images, SOPL technology can be utilized to create MNAs with a variety of structural designs [19]. Furthermore, compression test results indicate that MNAs produced using this method exhibit superior mechanical strength compared to those made with DLP printing. The technology allows for the accurate fabrication of complex MNA geometries. SOPL also enables quick customization, generating smooth surfaces that minimize insertion-related tissue damage and enhance biocompatibility [25].

One of the latest developments in 3D printing is 2PP, which offers nanoscale precision. This technique can fabricate MNAs with ultra-sharp tips and detailed architectures. While this technology is relatively expensive and time-consuming, its capacity to produce highly detailed structures makes it a valuable tool for research and niche applications [19,58].

2PP employs ultrashort laser pulses from a near-infrared femtosecond laser to selectively polymerize photosensitive resins. The process relies on the near-simultaneous absorption of two photons, producing an electronic excitation comparable to that of a single higher-energy photon. This results in a nonlinear energy distribution concentrated precisely at the laser’s focal point, with minimal exposure outside that zone. Within this focal volume, photoinitiators in the resin absorb the energy and trigger polymerization in localized regions known as “polymerization voxels”, where the energy surpasses a critical threshold. Compared to conventional techniques, 2PP offers unmatched control over geometry and resolution, while also lowering the infrastructure costs often associated with etching or lithography-based systems. Thanks to this precision and versatility, 2PP has been successfully used to fabricate both solid and hollow MNAs from a range of materials—including modified ceramics, hybrid inorganic–organic polymers, acrylate-based resins, polyethylene glycol, and, more recently, water-soluble compounds—with highly promising results [59].

A major advantage of 2PP is its ability to reach extremely fine resolutions—down to 100 nm [60]. Researchers have used this technique to shape dissolvable and hydrogel-forming MNAs from water-based mixtures of polyvinylpyrrolidone (PVP) and polyvinyl alcohol (PVA), enabling controlled drug delivery in skin model studies [59]. Additionally, 2PP has been applied to fabricate MNAs from organically modified ceramic hybrid materials (Ormocer^®^, Fraunhofer Institute for Silicate Research ISC, Würzburg, Germany) for transdermal drug administration [61]. MNAs produced using 2PP have demonstrated outstanding mechanical stability, showing no signs of breakage and little to no tip deformation even after surgical application [62]. Researchers have also developed a hybrid approach that combines 2PP with electrochemical deposition to fabricate ultra-sharp, gold-coated copper MNAs for targeted drug delivery to the inner ear [63]. Moreover, 2PP has been applied in the development of MNAs for transdermal sensing of electrolytes, including potassium (K^+^) ions [64].

FDM is another technique used for 3D printing of polymeric MNAs. It works by melting thermoplastic filaments and extruding them through a heated nozzle to construct structures layer by layer. Although FDM is cost-effective and easy to operate, it has limited resolution, which makes it less ideal for producing the fine, detailed features essential for MNAs [65,66]. Despite its limitations, recent improvements in FDM technology have made it usable for prototyping MNAs with simpler geometries. Also, while the resolution of FDM 3D printing is lower than that of other 3D printing techniques, post-processing methods such as chemical etching can enhance its effectiveness for MNA fabrication. This 3D printing technique is especially well-suited for producing biodegradable MNAs [67].

Several studies have used FDM to fabricate 3D-printed polymeric MNAs [68,69,70,71]. This 3D printing technique is widely favored because of its rapid fabrication, cost-effectiveness, accessibility, and versatility in material type [72]. However, post-processing plays a vital role in this technology, as the 3D-printed components are not immediately ready for use and post-processing is needed before using them [73]. FDM 3D-printed MNAs can penetrate the skin, break off, and deliver small molecules without requiring a master template or mold [68]. Additionally, coated polylactic acid (PLA) MNAs have been developed for effective transdermal drug delivery [71].

Table 1 summarizes the specifications of commonly used 3D printing technologies for fabricating MNAs, including SLA, DLP, LCD, SOPL, 2PP, and FDM. Also, Figure 2 shows the fundamentals of the main 3D printing processes used in the fabrication of polymeric MNAs. These include extrusion-based (FDM), projection-based (DLP, LCD, SOPL), and laser-based (SLA, 2PP) techniques. Variations in light sources and achievable resolutions are highlighted in the schematic.

### 2.2. Polymeric Material Used for 3D Printing of MNAs

The choice of material type in 3D-printed MNAs significantly impacts their mechanical properties, biocompatibility, and functional performance. This section explores a range of polymeric materials commonly used in 3D-printed MNA fabrication, including photopolymer resins, biodegradable polymers, hydrogels, composite resins, and emerging advanced polymers.

#### 2.2.1. Photopolymer Resins

Photopolymer resins are commonly employed in 3D printing because they solidify quickly when exposed to certain light wavelengths. Their ability to produce highly detailed and complex structures makes them particularly well-suited for fabricating MNAs [76]. Photopolymer resins are generally divided into UV-curable and visible light-curable types.

UV-curable resins have been widely applied in the 3D printing of MNAs [77,78,79], where exposure to UV light triggers photoinitiators to initiate polymerization. These resins are commonly used in SLA and DLP printing due to their ability to deliver high resolution and intricate detail, qualities that are critical for MNA fabrication. In addition to their fine structural precision, UV-curable resins also offer strong mechanical performance, ensuring that they can penetrate the tissue without fracturing. Their formulation can be tailored by incorporating various additives for curing and further improving desired properties [30,65]. Despite their advantages, a key challenge with UV-curable resins is ensuring their biocompatibility. Residual monomers and photoinitiators may pose cytotoxic risks, making it essential to carry out thorough post-processing to eliminate any unreacted substances [80].

Visible light-curable resins rely on light within the visible spectrum to trigger polymerization. They offer a key advantage in settings where gentler curing conditions are preferred over UV-based methods. These resins are generally considered safer and more convenient to handle, as they minimize the risks associated with UV radiation exposure [81]. These resins are also well-suited for certain polymers and allow the use of materials that could be damaged by UV exposure. However, despite their advantages, they can be limited by a shallower curing depth and generally lower mechanical strength compared to UV-curable alternatives [81].

#### 2.2.2. Biodegradable Polymers

Biodegradable polymers are beneficial for drug delivery applications where the MNAs dissolve after releasing their payload because they are made to break down gradually within the body. Among the most frequently utilized biodegradable polymers in 3D printing MNAs is PLA [71,82]. PLA is a well-liked material that can be processed using a variety of 3D printing techniques and is recognized for its biocompatibility and biodegradability. It offers MNAs sufficient mechanical strength. Another biodegradable polymer that breaks down more quickly than PLA is polyglycolic acid (PGA), which is frequently combined with other polymers to customize degradation rates [83,84]. Another biodegradable polymer, poly(lactic-co-glycolic acid) (PLGA), is a blend of PLA and PGA that provides mechanical characteristics and adjustable degradation rates. It is frequently utilized in drug delivery systems [85,86,87].

Removal after administration is not necessary when biodegradable polymers are used. However, it can be difficult to regulate the rate of degradation to meet the intended therapeutic need [88]. Additionally, it makes it possible for drugs to be encapsulated in the needle matrix and released into the tissue when it degrades or diffuses. For instance, studies have shown that MNAs derived from the biodegradable polymer poly(propylene fumarate-co-propylene succinate) (PPFPS) exhibit effective in vivo drug delivery properties [89].

#### 2.2.3. Hydrogels

Hydrogels are hydrophilic polymer networks capable of absorbing and retaining substantial amounts of water, making them highly suitable for controlled drug delivery applications [90]. Hydrogels are broadly classified into natural and synthetic categories. Natural hydrogels, such as alginate [91,92] and gelatin [93,94], are naturally biocompatible and can be chemically modified to enhance their functionality. In contrast, synthetic hydrogels like PEGDA [10,95] allow for precise tuning of mechanical and structural properties. Hydrogels are ideal for making MNAs that reduce insertion discomfort because of their softness and flexibility. However, generally speaking, they lack the mechanical strength of solid polymers, which could limit their usefulness in applications that call for deeper tissue penetration [96].

As previously stated, hydrogels’ special ability to absorb and retain water enables them to efficiently transport and release drugs gradually. Because of this, they are particularly helpful for transdermal drug delivery, providing consistent therapeutic effects with minimal discomfort. Additionally, their inherent body compatibility reduces the possibility of adverse effects, enhancing patient safety [90]. Hydrogels can react to environmental changes, such as changes in pH or glucose levels, and interact with biological molecules in the field of biosensing, allowing for precise, real-time tracking. Because of these characteristics, they are extremely valuable in advanced biomedical technologies [10].

According to research, 3D-printed gelatin GelMA hydrogel MNAs have tunable heights and sharp edges, providing remarkable mechanical performance without breaking at displacements of up to 0.3 mm. These results demonstrate that GelMA MNAs are mechanically strong, ideal for transdermal drug delivery, and offer high-resolution fabrication with precise control over geometry and mechanical characteristics [79].

#### 2.2.4. Composite Resins and Materials

Composite materials are formed by combining two or more different materials to minimize the disadvantages of each while maximizing the benefits of each. This frequently entails mixing polymers with either other polymers or inorganic components in 3D-printed MNAs [97,98]. By adjusting the MNAs’ mechanical strength, chemical behavior, and biological performance, researchers can make them suitable for a variety of applications. However, there are drawbacks to making composites; problems like uneven mixing and phase separation can make the fabrication process more difficult.

In a study [99], a novel 3D-printed MNA was fabricated from a composite hydrogel ink consisting of four monomers—2-(dimethylamino)ethyl methacrylate, N-isopropylacrylamide, acrylic acid, and acrylamide. The material, with superior mechanical characteristics, is ideal for high-precision 3D printing ad was produced by crosslinking with aluminum hydroxide nanoparticles. The resultant MNAs showed responsiveness to three stimuli—pH, temperature, and glucose—and had programmable shapes. The biocompatibility of the developed MNAs was further validated by their capacity to transdermally administer bovine serum albumin without causing cytotoxicity. This research demonstrated how sophisticated composite formulations can be used to fabricate multifunctional MNAs for use in biomedical applications.

#### 2.2.5. Stimuli Responsive Materials

New materials that may be helpful in 3D-printed MNAs are being steadily discovered through ongoing research. These innovative materials are made to perform better, be more biocompatible, and have new features. These include smart polymers, which are substances that can change how they behave in response to environmental changes such as variations in temperature, pH, or light exposure [100]. Conductive polymers [101] can be utilized for fabricating MNAs that not only deliver drugs but also have sensing capabilities in applications like biosensing. By incorporating nanoparticles into polymer matrices, nanocomposites can significantly improve the antibacterial qualities of MNAs, such as silver nanoparticles, which help prevent infections [102]. Novel materials have the potential to increase MNA technology’s capabilities, opening up new applications and enhancing current ones.

An exciting new area in intelligent drug delivery is the combination of AI-driven systems and stimuli-responsive materials. For instance, closed-loop insulin delivery can be supported by combining AI-enabled glucose monitoring systems with glucose-sensitive microneedles made of pH- or enzyme-responsive polymers. MNAs can adjust in real time to each patient’s physiological state by using machine learning algorithms to evaluate biosensor data and optimize actuation timing or dosage. These kinds of combinations might serve as the foundation for therapeutic platforms that are genuinely autonomous.

#### 2.2.6. Materials for 4D-Printed MNAs

The development of 4D printing in MNA fabrication requires the use of specialized materials. Unlike traditional 3D printing, 4D printing creates structures that can progressively change their shape, behavior, or properties in response to environmental stimuli like heat, moisture, or light. These changes occur because the materials are made to react and change in a specific, controlled way after printing. Time is the “fourth dimension” in this context because the printed object changes after it is first formed, allowing for dynamic, adaptive applications. In a study, a bioinspired MNA with backward-facing barbs for improved tissue adhesion was developed by Han et al. [95] using 4D printing. This technology made it possible to precisely create MNAs with intricate geometries, such as curved barbs, which are difficult to create with conventional techniques. These MNAs were created using projection microstereolithography (PµSL), which has programmable shape transformations that enable the barbs to curve backward when desolvated, greatly enhancing tissue adhesion. Figure 3 illustrates the 4D printing and post-processing workflow for their barbed MNAs schematically. The functionality of the MNAs was significantly influenced by the substance used in this procedure, particularly the photocurable resin (PEGDA 250). The barbs were able to bend and take on the required shape by varying the crosslinking density throughout the printing process. The creation of a highly effective MNA that exhibited 18 times stronger tissue adhesion than traditional MNAs was made possible by this carefully regulated material composition. Additionally, sustained delivery was demonstrated by drug release tests, underscoring the possibility of long-term uses in biosensing and drug delivery.

The type of material selected has a significant impact on the effectiveness, safety, and suitability of 3D-printed MNAs. Every alternative, including composites, biocompatible materials, photopolymer resins, and next-generation materials, has advantages and disadvantages of its own. A comprehensive understanding of these materials aids in the design of MNAs that meet the needs of particular medical applications. We can expect to see increasingly complex MNAs that expand their impact and utility in a range of clinical settings as 3D printing and materials science develop further.

## 3. Non-Transdermal Applications of Polymeric 3D-Printed MNAs

MNAs have emerged as a powerful technology for delivering therapeutics and monitoring biomarkers across various biological barriers beyond the skin. While early development of MNAs focused on transdermal patches, recent advances in materials and microfabrication (especially 3D printing) have expanded their applications to diverse non-transdermal targets [14]. Three-dimensional printing, including four-dimensional (4D) approaches using stimuli-responsive materials, enables the fabrication of MNAs with precise geometries, tunable mechanical properties, and complex architectures tailored to specific tissues. These 3D-printed MNAs can overcome key physiological barriers in organs such as the brain, eyes, oral cavity, GI tract, cardiovascular system, and reproductive tract, achieving localized drug delivery or diagnostic sampling with minimal invasiveness. Here, the latest developments in 3D-printed polymeric MNAs for non-transdermal drug delivery and diagnostic purposes, organized by target site, are discussed. In the discussed studies, the developed MNA may not be fabricated entirely by 3D printing, but 3D printing was utilized in the fabrication process. For each organ system, we discuss the unique challenges to drug delivery, how 3D-printed MNAs are designed to surmount those barriers, and specific examples from recent literature.

### 3.1. Brain/Central Nervous System (CNS)

Delivering drugs to the brain is notoriously difficult due to the blood–brain barrier (BBB) and the brain’s sensitive structure. The BBB’s tight endothelial junctions prevent most therapeutics from entering the brain parenchyma—it is estimated that over 98% of small-molecule drugs and essentially 100% of large biologics do not cross the BBB in sufficient amounts [103]. Systemic or oral administration often yields sub-therapeutic brain concentrations due to this barrier, while direct injection through the skull or intrathecal delivery is highly invasive and carries risks of tissue damage and infection. Precise targeting is crucial, as random diffusion can affect healthy brain regions. Thus, a major challenge is to achieve localized delivery past the BBB in a minimally invasive manner, avoiding systemic toxicity and bypassing the need for risky open-brain procedures.

Three-dimensional-printed MNAs offer a novel strategy to physically bypass the BBB and release drugs directly into the brain or cerebrospinal fluid compartments with high precision. By leveraging high-resolution 3D printing, MNAs can be made ultra-sharp and of microscale dimensions, enabling them to penetrate brain tissue or the meninges with minimal disruption [104]. For example, researchers have proposed integrating 3D-printed MNAs onto neural probes or catheters that, once inserted through a small cranial opening, can deliver therapeutics directly into a target region of the brain, thereby overcoming the permeability limitations of the BBB [105]. Importantly, 3D printing enables customization of needle geometry (length, tip profile, and barb features) to ensure adequate penetration of the soft brain tissue without causing hemorrhage or significant neuronal injury. Biocompatible polymer or hydrogel MNAs that are biodegradable can be used to leave behind drug depots in the brain that release over a sustained period, obviating the need for multiple injections. Researchers have also explored shape-memory or 4D-printed MNAs that can change configuration after insertion—for instance, swelling slightly to anchor in tissue or to modulate drug release kinetics. This is especially useful in the brain, where an MNA-based implant might need to remain in place and steadily release a drug (e.g., chemotherapeutics for a brain tumor or neurotrophic factors for regeneration) over days or weeks.

Although clinical examples are still in the early stages, recent reviews highlight the potential of 3D-printed MNAs in precision brain drug delivery [105]. One conceptual demonstration involved 3D printing an array of biodegradable polymer MNAs on a small implantable scaffold that could be positioned epidurally or intracerebrally to treat neurological disorders [105]. In preclinical studies, such MNA implants have been suggested for delivering chemotherapy drugs directly into brain tumors, achieving high local drug concentrations while sparing the rest of the brain and body [105].

Another emerging approach is transdural MNA delivery to the spinal cord: an MNA device can be placed on the dura mater (the membrane surrounding the spinal cord) to infuse drugs or nanoparticles into the cerebrospinal fluid for conditions like spinal cord injury or pain management [106]. By combining MNAs with a controlled release system (e.g., a metal–organic framework that gradually releases drugs), one study created a transdural patch that delivered therapeutics to the spinal cord over an extended period [106]. These examples underscore how MNAs can navigate the barriers of the CNS: physically breaching the BBB or dura in a localized fashion and then providing sustained, targeted drug administration directly to neural tissue. Although still largely experimental, the progress suggests that 3D-printed MNAs could enable new treatments for diseases like brain tumors, neurodegenerative disorders, and severe chronic pain—conditions for which the BBB has long limited therapeutic options. Future studies will focus on ensuring safety (e.g., no long-term tissue damage or inflammation in the CNS) and precise control of dosing, but the ability to customize MNA design via 3D printing provides an invaluable toolset for meeting these challenges.

A notable example of research on drug delivery to the brain is the silk fibroin-based MNA developed by Wang et al. [107], designed for glioblastoma (GBM) therapy. This heterogeneous silk MNA enabled spatially and temporally controlled release of multiple chemotherapeutic agents-thrombin, temozolomide (TMZ), and bevacizumab. These drugs were precisely loaded into different MNA regions using a multi-nozzle 3D printing strategy, allowing for sequential hemostasis, deoxyribonucleic acid (DNA) alkylation-induced apoptosis, and anti-angiogenic therapy. The patch showed improved drug stability, biocompatibility, and programmable release, including near-infrared-triggered delivery of bevacizumab, resulting in significant tumor reduction and prolonged survival in GBM-bearing mice.

One of the main goals has been to develop materials that can pass through the BBB in order to treat brain tumors. However, the BBB also makes it difficult for chemotherapeutic medications to enter the brain tissue and treat brain disorders [108,109]. In order to overcome these obstacles, Lee et al. [110] designed thin MNs that allow trypan blue dye to be deeply injected into a mouse model’s brain. The developed MNs had dimensions of 5.3 mm in length, 40 μm in thickness, and 70 μm in width.

Efforts have been made to address the challenge of delivering anti-tumor drugs to brain tumors [111]. In a study, Wang et al. [112] developed a heterogeneous and multifunctional silk MNA device for the in situ treatment of brain glioma. The MNAs were fabricated using a silk fibroin matrix combined with inkjet printing to precisely load various drugs—such as tumor inhibitors, antiangiogenic agents, and hemostatic agents—into specific MNA regions. This allows for spatial and temporal control of drug release. The device, designed to be flexible and biodegradable, conforms to the curved surface of the brain and degrades after delivering its payload. Drug release rates can be tuned by altering the crystallinity of silk via ethanol treatment. In vitro and in vivo experiments demonstrated controlled and sustained drug release, effective penetration of brain tissue, and significant inhibition of tumor cell viability. This innovation circumvents the blood–brain barrier and provides a customizable, localized therapeutic strategy for postoperative glioma management [112].

For localized glioma therapy, Wang et al. [113] developed a silk-based MNA that can deliver several medications with different release profiles. The developed MNA was implanted, and even at lower dosages, it showed improved therapeutic efficacy in a mouse model with gliomas. The outcomes revealed that this MNA had a great deal of promise as a novel glioma treatment approach. The device made it possible to deliver medications directly into the tumor cavity with controlled, variable release kinetics by circumventing the blood–brain barrier. The system successfully inhibited the growth of glioma cells in the tumor model, increasing survival rates and improving treatment results. To fabricate the device, three therapeutic agents—temozolomide, bevacizumab, and thrombin—were blended with a silk protein solution to produce a functionalized silk ink, which was then injected into separate mold compartments to form the final MNA structure [113].

Another study by Muresan et al. [114] created polymeric MNAs that are meant to be implanted into the resection cavity after tumors like isocitrate dehydrogenase wild-type glioblastoma are surgically removed. These MNAs were loaded with polymer-coated nanoparticles (NPs) that contained either olaparib or cannabidiol. Drug release and diffusion were assessed using ex vivo rat brain tissue and an in vitro brain model. To simulate brain tissue, methylene blue-loaded MNAs were inserted into a 0.6% agarose gel cavity, where they formed visible channels that enabled lateral dye diffusion. When cannabidiol-loaded NPs were used, the agarose exhibited a concentration of 12.5 µg/g at 0.5 cm from the MNA insertion point. High-performance liquid chromatography further confirmed the successful delivery of cannabidiol at 59.6 µg/g into ex vivo brain tissue. Likewise, olaparib-loaded MNAs delivered 5.2 µg/g of the drug into agarose gel at the same distance. Both olaparib and MNA material were confirmed to be present in the rat brain hemisphere up to 6 mm from the insertion site using Orbitrap secondary ion mass spectrometry. These results support continued research on MNAs in neuro-oncological applications and demonstrate their promising potential for localized drug delivery within the brain [114].

In a study on brain-targeted drug delivery, Sarker et al. [17] introduced a novel hybrid additive manufacturing technique that combines ex situ direct laser writing with DLP 3D printing to create advanced MNAs for fluidic microinjections. The manufactured MNAs’ hollow structures measured 550 µm in height, 30 µm in inner diameter, 50 µm in outer diameter, and 100 µm in needle spacing. During microfluidic cyclic burst-pressure testing, the DLP-printed capillaries maintained their fluidic and structural integrity at the MNA–capillary interface even after being exposed to pressures greater than 250 kPa for 100 cycles. Ex vivo experiments on mouse brain tissue demonstrated that the MNAs could successfully penetrate and retract from the brain surface, while also enabling uniform and efficient microinjection of surrogate fluids and nanoparticle solutions. Overall, the study highlights the strong potential of this high-resolution, high-density hollow MNA fabrication strategy for a range of biomedical microinjection applications. Figure 4 shows their developed MNA system designed for brain drug delivery, including its schematic, morphological characterization, experimental setup, and demonstration of the injection process [17].

Overall, 3D-printed polymeric MNAs represent a promising platform for brain/CNS applications, offering advantages such as high customization, on-demand release, and reduced systemic toxicity. As demonstrated by these studies, this technology is paving the way for novel treatments of brain tumors, neurodegenerative diseases, and CNS injuries.

Despite encouraging preclinical results, it should be highlighted that most of the studies discussed in this section for the brain/CNS drug delivery and diagnostic application are still in the proof-of-concept stage and mostly use animal models or simulated brain environments. These models fall short in capturing the complexity of human neuroanatomy, immunological responses, and tissue healing dynamics. Sample sizes are usually small, and little is known about the long-term biocompatibility, degradation, and potential neurotoxicity of implanted polymers. The absence of large-animal or early-phase clinical data significantly restricts our understanding of how these technologies will work in humans.

### 3.2. Oral Cavity

The oral cavity (including buccal mucosa, gums, tongue, etc.) presents a unique set of barriers for drug delivery. The oral mucosal epithelium is thinner than skin but still limits the permeation of many drugs, especially large or hydrophilic molecules. Saliva constantly wets the tissues, which can rapidly wash away drug formulations and dilute them [11]. Enzymes in saliva may degrade drugs, and swallowing can lead to unintentional systemic absorption or loss of the drug to the GI tract. Moreover, delivering drugs to localized oral lesions (such as oral cancers or periodontal disease sites) is challenging with systemic therapy due to limited blood supply in some regions and the tendency of drugs to diffuse away. In dental applications (e.g., treating periodontal pockets or oral ulcers), achieving sufficient local drug concentration without frequent reapplication is difficult. Standard injections in the oral cavity (for local anesthesia or chemotherapy into a tumor) can be painful and imprecise, causing tissue damage and uneven drug distribution [11]. Thus, the oral cavity’s constant fluid flow and delicate tissues demand a delivery system that can securely adhere and concentrate drugs at the target site while avoiding patient discomfort.

MNAs provide a minimally invasive means to penetrate the superficial oral mucosa or even intraoral tumors to deliver drugs directly where needed. Using 3D printing, MNAs can be fabricated from biocompatible polymers or hydrogels that are safe for oral use and can be made dissolvable or swellable. This allows the needles to release their payload and then harmlessly degrade in the mouth without leaving sharp residues. Crucially, 3D printing enables the incorporation of design features to address the wet environment of the mouth: for instance, MNAs can include mucoadhesive backing layers or barbs that help them latch onto the slick oral mucosa despite saliva flow [11]. High-resolution 3D printing also yields very sharp needle tips that penetrate the surface quickly and painlessly (avoiding the rich network of nerves that cause injection pain). Once inserted, MNs form microchannels that bypass the barrier epithelium, facilitating rapid drug entry into the tissue. Additionally, 4D-printable materials can be used for oral MNAs that respond to saliva or pH by expanding and becoming more adhesive, thereby counteracting the washing effect of saliva. For example, a 3D-printed MNA made of swellable chitosan-based hydrogel could adhere strongly to the moist gum tissue and slowly release an antibiotic or growth factor to treat periodontal disease over hours to days.

In a study, He et al. [75] developed a minimally invasive therapeutic system using 3D-printed hydrogel MNAs loaded with madecassoside to promote periodontal soft tissue regeneration. Using DLP 3D printing, the authors fabricated PEGDA-based MNAs with excellent mechanical properties, biocompatibility, and drug release capability. In vitro experiments showed that madecassoside enhanced the proliferation of human gingival fibroblasts and increased type I collagen expression. Animal studies confirmed significant improvements in gingival height, thickness, and collagen deposition in treated rabbits. The MNA system successfully delivered the therapeutic compound trans-mucosally, offering a non-surgical alternative for treating gingival recession. This platform demonstrates great potential for future periodontal and oral mucosa therapies due to its precision, efficacy, and low invasiveness. As shown in Figure 5, the fabricated hydrogel MNA exhibited well-defined morphology and structural integrity (A–C). The system demonstrates a sustained drug release profile over 6 h (D) and maintains good biocompatibility, with cell viability remaining above the cytotoxicity threshold during the testing period (E) [75].

In a periodontal wound model, the MNAs could insert into the gingival tissue around teeth and locally release madecassoside, significantly accelerating the healing of the gums [75,115,116,117]. Unlike traditional mouth rinses or gels that would be washed away by saliva, the MNAs ensured that the drug was delivered *into* the tissue and retained at the site. Another application is treating oral cancer.

Collectively, the studies highlight the versatility and efficacy of 3D-printed polymeric MNAs in oral cavity drug delivery and diagnostics.

### 3.3. Ocular (Eye)

The eye is a highly protected organ with multiple barriers that make drug delivery difficult. The cornea and sclera (the outer layers of the eye) guard against entry of foreign substances; topical eye drops often penetrate only a tiny fraction of the dose into the eye. The blood–ocular barriers (the blood–aqueous and blood–retinal barriers) function similarly to the BBB in preventing systemic drugs from reaching intraocular tissues. Treating diseases at the back of the eye (retina or choroid) typically requires repeated intravitreal injections (needle injections into the vitreous humor), which carry risks of infection, retinal detachment, and patient discomfort. Even delivering drugs to the anterior segment (cornea, anterior chamber) is challenging—eyedrops are washed away by tears and the blink reflex within minutes, and less than 5% of the drug in eye drops actually penetrates the cornea. Achieving sustained therapeutic levels in the eye thus usually demands either very frequent dosing or implanting devices, both of which have drawbacks. An ideal ocular delivery method would bypass the external barriers and accurately deposit the drug at the required site (cornea, retina, etc.) with minimal invasiveness.

MNAs provide a compelling approach for ocular drug delivery by creating micron-scale pathways into specific eye tissues in a targeted, minimally invasive way. Using 3D printing, engineers can create MNs short enough to avoid perforating the entire cornea or sclera but long enough to deposit drugs at the required depth (for example, within the corneal stroma or the suprachoroidal space between sclera and choroid) [11]. The fine control of needle geometry afforded by 3D printing is crucial here—ocular tissues are only a few hundred microns thick in some cases, so precision is needed to avoid overshooting the target. DLP and 2PP have been used to fabricate MNAs with tip lengths on the order of tens to a few hundred microns, with ultra-sharp tips to painlessly penetrate the eye’s surface [11]. Three-dimensional-printed applicator devices are also being utilized alongside MNAs to improve delivery. A prime example is the spring-loaded MNA pen designed for corneal injection [11]. The pen houses a single MN (which can be 3D-printed or made by molding a printed master) and uses a calibrated spring mechanism to swiftly insert the MN into the cornea to a defined depth. This approach compensates for the cornea’s flexibility and the lack of counter-pressure (since the cornea is thin and backed by fluid), achieving reliable penetration where manual insertion might fail [11]. Additionally, 3D printing allows the incorporation of features like curved or concave baseplates so that an MNA can conform to the globe of the eye. Special MN designs with barbs or tilted tips have been explored for the eye as well, aiming to lock into the tissue and resist the blinking or movements that could dislodge a patch. Dissolvable MNAs made of ocular-friendly polymers (e.g., hyaluronic acid, methacrylated gelatin) can release drugs like anti-VEGF biologics or steroids over time and then disappear, avoiding the need for device retrieval from the eye. Overall, additive manufacturing allows the fabrication of MNA systems that are finely tuned for the eye’s delicate anatomy, improving the safety and effectiveness of ocular drug delivery.

The aforementioned advantages of 3D printing facilitate the development of polymeric MNAs with various geometries and drug-loading profiles that are appropriate for specific ocular conditions such as keratitis, glaucoma, and diabetic retinopathy. These devices offer sustained drug release over days to weeks, reducing dosing frequency and improving patient compliance—an essential factor in chronic ocular conditions where adherence to treatment regimens is low.

Gade et al. [118] explored the use of hollow MNAs for localized intrascleral drug delivery, emphasizing the integration of polymeric 3D-printed adapters to enhance precision and reproducibility. The study addressed the challenges of posterior eye segment drug targeting by developing hollow MNAs in combination with custom-designed adapters that controlled both injection angle and volume. Using porcine eye models, they demonstrated that a 60° bevel angle, paired with a 45° injection orientation guided by the 3D-printed adapter, enabled efficient drug depot formation within the scleral layers, with minimal tissue disruption. The adapters, fabricated from biocompatible photopolymer resins, allowed accurate microinjections of rhodamine B dye, confirmed via histology and multiphoton microscopy. The study showed that such polymeric 3D-printed components can play a critical role in ensuring controlled, localized delivery in ocular applications. This approach highlights the potential of integrating polymeric micromanufacturing tools with MNA systems for non-transdermal drug delivery.

Fitaihi et al. [37] developed and optimized an SLA 3D printing process to fabricate MNA micro-molds for ocular drug delivery applications. They systematically investigated the influence of key parameters—including MN shape, aspect ratio, layer thickness, and printing orientation—on mold fidelity and needle sharpness. Three geometric shapes (cone, pyramid, and triangular pyramid) with varying aspect ratios were evaluated, with cone-shaped MNs at a 1:2 aspect ratio yielding the best balance between mechanical strength and printing precision. The optimized MNs were used to fabricate polymeric patches from PVP/PVA blends. Mechanical and insertion studies demonstrated that MNs printed at a 67.5° orientation with 25 μm layer thickness provided the sharpest tips and effective penetration into porcine corneal and scleral tissues with minimal force. The study highlights the potential of low-cost, customizable SLA printing for producing patient-friendly ocular MN patches.

In conclusion, 3D-printed polymeric MNAs are revolutionizing ocular drug delivery by combining minimally invasive insertion, controlled drug release, and patient comfort. As fabrication technologies and biopolymer formulations advance, the translation of these devices into clinical practice for ophthalmic therapies and diagnostics appears increasingly feasible.

### 3.4. Gastrointestinal Tract

The GI tract, especially the stomach and intestines, is an extremely harsh environment for drug delivery. Orally administered drugs must survive acidic pH in the stomach, degradative enzymes throughout the GI tract, and first-pass metabolism in the liver. Many biologic drugs (insulin, vaccines, antibodies) are ineffective orally because they are destroyed before absorption. Even for drugs that are absorbed, GI transit can be unpredictable, and only a small fraction of the dose might reach systemic circulation. Moreover, some therapies would benefit from localized delivery in the GI tract (for example, treating inflammatory bowel disease or GI cancers) without systemic exposure, but delivering drugs to a specific spot in the intestine and keeping them there long enough to act is difficult with traditional capsules or injections. Conventional oral dosage forms also cannot easily target the intestinal lymphatics or the portal vein specifically. Swallowable electronics or capsules can provide new capabilities, but ensuring that they reliably attach to or inject into the GI mucosa is challenging, given peristalsis and the mucus lining.

Three-dimensional printing has enabled the design of ingestible MNA pills or capsules that contain MNAs on their surface, which can mechanically insert into the GI mucosa (stomach or intestinal wall) to deliver drugs directly into the tissue or bloodstream. Unlike a normal pill that releases a drug into the GI lumen (where much is degraded or excreted), an MNA-equipped capsule can inject the drug into the wall of the stomach or intestine, bypassing the harsh lumen environment [11]. Additive manufacturing is crucial in this context for rapidly prototyping capsules with complex moving parts and precisely positioned needles. For example, researchers have 3D-printed capsule shells and internal springs that can deploy MNAs on command; one design uses a magnetically triggered mechanism in which an external magnet causes the capsule to open its MNA “arms” and press into the intestinal wall [119]. The capsule and its cantilever-based injection module were entirely 3D-printed using a photopolymer resin, demonstrating how printing can integrate drug reservoirs, needles, and actuating components in a single ingestible device [119]. Capsules can be shaped using 3D printing to increase the likelihood that they will correctly align and make contact with the epithelium (for example, a self-righting shape that consistently lands needle-side down on the gut wall). Some designs include dissolvable stabilizing structures that keep the MNA folded until the capsule reaches a target GI region (such as a specific pH in the intestine), at which point the structure dissolves and releases a spring that pushes the needles out. This level of sophistication in design is made feasible by modern 3D printers.

A pioneering example of an MNA pill that demonstrated improved oral delivery of insulin. In a study, Abramson et al. [120] developed an innovative ingestible device called the luminal unfolding MNA injector (LUMI) for the oral delivery of biologic macromolecules, such as insulin. The capsule-based system is designed to deploy in the small intestine, where it unfolds to insert dissolvable, drug-loaded MNAs into the intestinal tissue. This approach bypasses the barriers of enzymatic degradation and poor absorption associated with conventional oral drug delivery. The team engineered LUMI with biodegradable arms and a spring-loaded core to optimize tissue contact and MNA penetration without causing perforation. The system was tested in ex vivo human and in vivo swine models, demonstrating safe deployment, robust tissue engagement, and significant systemic uptake. Insulin-loaded LUMI devices induced a measurable reduction in blood glucose, achieving over 10% bioavailability relative to subcutaneous injection. This work suggests a promising platform for non-invasive systemic delivery of biologics currently limited to injectable formulations.

Building on such concepts, newer devices have incorporated 3D-printed parts for better functionality. Recently, a magnetically triggered ingestible capsule was developed with fully 3D-printed housing and a reservoir for liquid drugs. Levy et al. [119] developed a magnetically triggered ingestible capsule designed for localized drug delivery within the GI tract using dissolvable MNAs. This capsule houses a flexible PEEK cantilever embedded with an MNA, which is held in place by a low-melting EVA adhesive. Upon exposure to an external magnetic field, a reed switch activates a resistive heater that melts the adhesive, releasing the cantilever and inserting the drug-loaded MNAs into the intestinal tissue in under 3 s. The PEEK cantilever demonstrated mechanical stability, rapid deployment, and minimal stress relaxation, making it ideal for GI applications. The system achieved successful deployment in ex vivo porcine models, showing enhanced localization and drug diffusion compared to conventional methods. The capsule’s modular design, low power requirements, and scalability indicate strong potential for clinical translation in targeted GI drug therapies without systemic side effects. Figure 6 depicts this developed MNA system.

In another approach, 3D-printed microcontainers with dissolving MNAs have been explored for vaccines: the capsule protects the vaccine through the stomach, and then in the intestines, the capsule opens and presses a dissolvable MNA into the intestinal mucosa to administer the vaccine to the rich network of immune cells there (Peyer’s patches) [121]. Beyond drug delivery, 3D-printed MNA capsules can also serve diagnostic purposes. Researchers at Purdue University created a 3D-printed capsule designed to sample gut bacteria by exposing MNAs that “harpoon” microorganisms in the stomach [122]. This capsule was capable of capturing microbiome samples from the GI tract for analysis, offering a non-invasive way to diagnose infections or gut flora imbalances that would normally require an endoscopic procedure [122]. These innovations illustrate how 3D printing empowers complex capsule designs that can actively interact with GI tissue. As this field progresses, we may see patient-specific printable smart pills that deliver drugs at precise GI locations or continuously monitor GI biomarkers—all enabled by MNAs and advanced manufacturing. One important consideration is safety: robust testing is needed to ensure that MNA capsules do not cause perforations or ulcerations in the GI lining. Thus far, the prototypes (with careful control of needle length and use of self-disabling features after drug release) have been well-tolerated in animal models [11].

Building upon the concept of localized insertion, Chen et al. [123] developed a dynamic omnidirectional adhesive microneedle system (DOAMS) tailored for oral delivery of macromolecules. Although primarily investigated in the upper GI tract, these adhesive MNAs—constructed from 3D-printed stimuli-responsive polymers—demonstrated the ability to anchor onto moist mucosal surfaces, adapt to motility-induced shear forces, and enable robust interfacial drug transfer. These systems utilize MNAs with tunable stiffness and biodegradable backings, facilitating sustained release and minimizing mucosal irritation. The DOAMS system was particularly effective in delivering therapeutic proteins, such as insulin and exenatide, and achieved significant glycemic control in animal models. Figure 7 illustrates the design and in vivo performance of the DOAMS-based MNA system for gastric drug delivery. The schematic (A) shows how the capsule deploys in the stomach and anchors to tissue for localized release. SEM imaging (B) confirms the structure of the MNAs. In vivo studies in pigs (C–E) demonstrate significantly improved semaglutide absorption with DOAMS tablets compared to controls.

These examples illustrate how 3D-printed MNAs are addressing critical limitations of conventional oral drug delivery systems. By enabling the physical insertion of therapeutic agents across mucosal barriers, such devices provide an alternative to injections for chronic therapies like diabetes, potentially improving patient adherence and therapeutic outcomes.

Overall, 3D-printed polymeric MNAs for GI tract applications offer a promising pathway to transform the oral administration of biologics. Future developments may extend their utility to GI diagnostics through embedded biosensors or responsive drug delivery for conditions like Crohn’s disease or GI cancers.

The translational pathway remains unclear despite the promising performance of 3D-printed MNA capsules in early in vivo studies and ex vivo animal models. Several systems under discussion are still in the early phases of development and have not yet been proven to work on humans or in larger animal models. Additionally, issues like mechanical safety under peristaltic forces, mucosal immune responses to embedded polymers, and long-term tissue interactions are rarely addressed. As a result, caution should be used when applying recent research to clinical settings.

### 3.5. Cardiovascular System

Reaching targets within the cardiovascular system—including heart tissue and blood vessels—involves overcoming dynamic physiological factors. The heart is constantly contracting and is encased in the pericardium, making it difficult to maintain contact with a drug delivery device. The myocardium (heart muscle) has a high blood flow that can wash drugs away quickly, and systemic delivery of cardiovascular drugs often dilutes the drug or causes off-target effects. For example, delivering regenerative factors or cells to a heart after myocardial infarction is challenging; intracoronary infusions result in most of the therapeutic agents being carried away by blood flow, and direct myocardial injections with needles often have poor retention (the “leak out” problem) and can cause local tissue damage. Blood vessels themselves are difficult to treat locally; conditions like restenosis or atherosclerotic plaques are typically treated with systemic drugs or catheter-based interventions. A major challenge is achieving site-specific deposition of therapeutics in the cardiovascular system that remains in place despite blood flow or cardiac motion. Any device must also be deployable via minimally invasive means (e.g., through catheters) because open-chest surgery is high-risk.

MNA technologies are being applied in two main contexts in the cardiovascular system: (1) patches or devices applied to the heart for localized delivery and (2) endovascular or perivascular devices for blood vessels. In both cases, 3D printing offers the ability to create complex, customized MNA geometries that address mechanical challenges. For cardiac applications, researchers have developed 3D-printed MNAs that can be placed onto the epicardium (outer surface of the heart). These patches often incorporate *self-anchoring features*, such as backward-facing barbs or bendable needles, to ensure that the patch stays attached to the beating heart. A recent design involved a 3D-printed polymer patch with MNAs containing tiny hooks—upon insertion into the heart muscle, the MNAs latch on like fishhooks, preventing the patch from detaching during systole and diastole. Such a design was impossible to manufacture with traditional molding, but high-resolution 3D printing allowed the inclusion of these intricate barbs on each 500 µm diameter needle. Furthermore, 3D printing enables MNAs to have tunable mechanical properties (e.g., a flexible base that conforms to heart curvature but rigid needles to penetrate the epicardium). For vascular applications, additive manufacturing can produce MNA-equipped catheter tips or balloon catheters. One example is akin to the Bullfrog^®^ micro-infusion device: a balloon catheter that, when inflated, pushes out a small needle to inject the drug into the vessel wall [11]. With 3D printing, multiple MNs could be integrated on a balloon’s surface to create an array that uniformly injects around the circumference of an artery. Another concept is a perivascular wrap—a 3D-printed flexible strip with MNs that can be wrapped around a blood vessel during surgery to deliver anti-proliferative drugs directly to an arterial lesion (for preventing restenosis) [11]. In short, the design freedom of 3D printing allows cardiovascular MNAs to incorporate features like targeted deployment (through catheters) and secure attachment to moving tissues, which are essential for success in the circulatory system.

Recent studies have demonstrated the feasibility of MNA approaches in cardiovascular therapy. Advanced designs like self-interlocking MNAs for cardiac use—for instance, a patch with center-facing tilted needles that clamps onto cardiac tissue and a patch with barbed needles that hook into the myocardium, both of which have been shown to maintain secure contact during the cardiac cycle. These design innovations, made possible by precise micro-fabrication (e.g., 3D printing or laser lithography), ensure that the therapeutic patch remains in place long enough to deliver its payload despite the heart’s motion [124].

In the vascular realm, one commercially inspired device (Bullfrog^®^, Mercator MedSystems Inc., San Leandro, CA, USA) combines a micro-needle with a balloon catheter to deliver drugs into blood vessel walls. Similar catheter–MN systems have been adapted for other tubular tissues as well [11]. A research example for peripheral artery disease is using an MN catheter to inject anti-proliferative drugs into the arterial wall at the site of a stent or plaque to prevent scar tissue formation. Three-dimensional printing can enhance these by enabling multi-needle arrays on a single inflatable platform, thus treating a larger area uniformly. Another interesting approach is using dissolvable MNA plugs to treat vascular malformations: one could inject an MNA that leaves behind a polymer depot inside a vessel wall that slowly releases a drug or blocks an abnormal vessel (like a targeted embolization). While such an application is still conceptual, it takes advantage of the combination of mechanical action and chemical delivery that MNAs provide. Diagnostic uses in the cardiovascular system are also being explored. For example, an MNA sensor could be inserted into a vessel to continuously measure blood analytes (glucose, electrolytes) or detect inflammatory markers. Researchers have 3D-printed polymer MNAs coated with enzymatic sensing layers, intended for insertion into tissue to sample ISF; a similar principle could be applied to superficial arteries or veins for short-term monitoring. In summary, 3D-printed MNAs are pushing into cardiovascular applications that were once thought impractical for such devices, from helping damaged hearts heal themselves by precise cellular therapy delivery to enabling localized drug administration within blood vessels. As these technologies progress, they might reduce the need for systemic drugs (and their side effects) in cardiovascular disease by offering targeted, high-concentration therapy exactly where it is needed.

Polymeric MNAs could be used for treating myocardial infarction by delivering regenerative cells, bioactive molecules, and drugs directly to damaged heart tissue. These MNAs provide mechanical support and sustained localized release, overcoming the limitations of systemic therapies and stents [125].

These studies exemplify how 3D-printed polymeric MNAs can be precisely engineered for targeted therapeutic delivery and regenerative medicine in cardiovascular care. The ability to fabricate MNAs with tunable mechanical and release properties through 3D printing technologies enables customization for specific clinical needs. Moreover, the integration of cell therapy with MNA platforms heralds a new era of minimally invasive cardiac repair strategies. As this field advances, future work should focus on optimizing materials for biodegradability, refining delivery systems for minimally invasive deployment, and extending the clinical applicability of MNAs to broader cardiovascular disorders.

### 3.6. Reproductive System

Drug delivery to reproductive organs (such as the vaginal mucosa, cervix, uterus, or male reproductive tract) often requires surmounting mucosal barriers and achieving localized effects while minimizing systemic uptake. The vaginal cavity, for instance, has a large surface area and is an attractive route for vaccines or medications (e.g., for sexually transmitted infection prevention or local hormone delivery). However, conventional vaginal formulations like gels or rings sometimes suffer from uneven distribution and leakage. The vaginal epithelium, while permeable to some extent, still limits large molecule transport, and there is a constant turnover of mucus. Delivering drugs to the uterus (for conditions like endometrial disease) or to ovarian tissue is even more challenging without invasive procedures [126]. Similarly, in the male reproductive system, targeting drugs to the testes or prostate is difficult due to tissue barriers (like the blood–testis barrier). Local injection with needles can be painful and carries the risk of infection in these sensitive areas. Thus, a major challenge is providing sustained, localized drug delivery or vaccination in reproductive tissues in a way that is pain-free, user-friendly, and effective despite the protective barriers.

MNAs provide a promising solution for delivering drugs and vaccines to the reproductive tract mucosa in a minimally invasive manner. A dissolvable MNA applied to the vaginal wall can rapidly bypass the superficial mucus layer and deposit vaccine antigens or drugs in the underlying tissue, achieving higher bioavailability than topical gels. Three-dimensional printing plays an enabling role by allowing the creation of MNA shapes optimized for mucosal adhesion and retention. For example, a 3D-printed MNA might have a wider base or a suction cup-like concave base that presses against the moist tissue to stay in place. The materials used can be tuned: 3D-printed hydrogel MNAs that swell upon contact with vaginal fluids will not only release the drug but also help stick the patch to the tissue (acting like an adhesive). The gentle, pain-free nature of MNA insertion is particularly suitable for vaccines targeting the reproductive tract, where traditional injections might deter patient compliance. With 3D printing, one can also produce patient-specific devices; for instance, an anatomically contoured MNA that fits the shape of the cervical tissue could be printed to ensure uniform contact, something not feasible with one-size-fits-all molds. Additionally, 4D printing (using stimuli-responsive polymers) could be employed so that a vaginal MNA only exposes its needles when it senses the correct environment (pH or temperature), adding a layer of safety and comfort during insertion. For male applications, while less reported, one could imagine a 3D-printed MNA system to deliver contraceptives or anti-inflammatory drugs to the testes or prostate in a targeted way, perhaps via a minimally invasive applicator that avoids systemic effects.

For mucosal vaccination using MNAs, Wang et al. [127] developed a dissolvable MNA for vaginal delivery of a human immunodeficiency virus protein vaccine and adjuvant. The MNAs were made of sugars and polymers (sucrose, PVP, carboxymethyl cellulose (CMC)) formulated to preserve the stability of the protein antigens. These MNAs were molded using a polydimethylsiloxane (PDMS) template (itself created from a master—a process that could utilize a 3D-printed master mold). When applied to the vaginal walls of mice, the dissolving MNAs delivered the vaccine into the tissue, inducing both local and systemic immune responses. The study reported that the MNA-delivered vaccine achieved effective immunization without any observable tissue damage or safety issues, highlighting the approach’s promise for sexually transmitted infection vaccines.

In summary, 3D-printed polymeric MNAs, particularly those with bioinspired geometries and multifunctional material composition, hold significant potential in reproductive healthcare. The work of Zhang et al. [126] exemplifies how advanced MNA technologies can be engineered to address complex gynecological disorders through targeted delivery, mechanical stability, and tissue specificity—ushering in a new era of minimally invasive therapies for female reproductive system disorders.

It should be highlighted that the majority of studies are restricted to small animal models or in vitro experimental setups, despite the fact that research and studies have demonstrated encouraging results for local drug and vaccine delivery and detection using MNA systems in reproductive tissues. These results are preliminary, and little is known about the effects on human tissue, especially with respect to biodegradation, chronic exposure, and mucosal sensitivity. There are currently no established regulatory standards for microneedle systems designed specifically for reproductive use, and there are few long-term safety and effectiveness data available.

### 3.7. Other Emerging Areas

Beyond the major organ systems discussed above, there are several emerging and niche applications of 3D-printed MNAs that highlight the versatility of this technology. Here, we discuss drug delivery to the inner ear, targeted treatment of specific glands or tissues, and innovative diagnostic uses.

#### 3.7.1. Inner Ear

The inner ear (the cochlea and vestibular system) is separated from the middle ear by the round window membrane (RWM), a delicate barrier similar to the BBB in its protective function [128]. Delivering drugs to treat hearing loss or balance disorders has been extremely challenging—systemic drugs do not effectively reach the cochlea, and intratympanic injections (placing the drug in the middle ear) rely on slow diffusion across the RWM. Recognizing this, researchers have developed 3D-printed MNAs for inner ear delivery. Chiang et al. [129] developed and evaluated two designs of 3D-printed MNAs intended to perforate the human round window membrane (HRWM) precisely and safely to facilitate drug and diagnostic delivery into the inner ear. Using 2PP, MNs with diameters of 100 μm and 150 μm were fabricated and tested ex vivo on human temporal bones. Eighteen perforations were performed, and force–displacement data were recorded. Confocal microscopy revealed that the MNAs created slit-shaped perforations aligned with collagen fibers, with major axes closely matching the MN diameters. Minimal MNA displacement was required for complete perforation, ensuring safety from intracochlear damage. SEM showed the MNAs remained structurally intact after use. Peak perforation forces ranged from 46–71 mN, and displacement remained well below the threshold that could risk damaging the cochlea. This study demonstrates that 3D-printed MNAs can reproducibly and accurately perforate the HRWM, paving the way for safer, minimally invasive inner ear therapeutic strategies. The team developed this device, and it is now commercially available [128,130]. The MNA, printed via 2PP, has a tip radius of less than 1% of a human hair’s thickness, enabling it to push aside the membrane’s fibers rather than tear them [128]. In guinea pig studies, a single MN perforation in the RWM allowed drugs placed in the middle ear to rapidly diffuse into the inner ear, achieving effective inner ear drug levels [128]. Remarkably, the tiny hole began to heal within 48 h and fully closed in one week, and hearing tests showed that hearing returned to normal within a day after MNA use [128]. This demonstrates the MNA’s precision in creating a controlled, reversible opening. By delivering drugs such as steroids or gene therapy vectors directly into the cochlea, this approach could vastly improve treatments for sudden hearing loss, ototoxicity, or inner ear infections. The MNA can also potentially be made hollow to inject drugs or even withdraw fluid from the cochlea for diagnostics [128]. Indeed, one report notes that this 3D-printed needle allows for cochlear fluid removal to diagnose conditions like Meniere’s disease [129]. The inner ear MNA exemplifies how 3D printing enables extremely fine, custom-shaped needles that solve a previously intractable drug delivery problem.

#### 3.7.2. Targeted Organ Delivery and Tumor Therapy

Outside of the standard routes, researchers are exploring 3D-printed MNAs for organ-specific interventions. For instance, MNA devices have been considered for lymph node targeting—a 3D-printed MNA could be inserted into a lymph node during surgery to slowly release immunotherapeutics or to serve as a biosensor for metastatic cancer cells. Similarly, there is interest in using MNAs for intratumoral drug delivery in accessible tumors (e.g., breast or head-and-neck tumors). A 3D-printed array of biodegradable needles could be implanted into a solid tumor to provide high local chemo- or immunotherapy, avoiding systemic toxicity. While Lee et al. (2020) already demonstrated this concept in oral tumors with etched metal MNAs [11], 3D printing could allow customized needle lengths and spacings depending on tumor size and location, potentially improving distribution within heterogeneous tumor tissue. Another emerging area is organ-on-chip integration: 3D-printed MNAs can interface with microfluidic organs-on-chips for research, delivering substances to miniature tissue constructs or sampling cell secretions in real time for analysis [75]. This is more of a diagnostic/research tool, but it leverages the same principles.

#### 3.7.3. Point-of-Care 3D Printing of MNAs

A final emerging trend worth noting is the advent of on-demand 3D printing of MNAs for rapid response situations. Vander Straeten et al. [131] developed a microneedle vaccine printer (MVP) capable of producing dissolvable MNAs loaded with lipid nanoparticle (LNP)-encapsulated messenger ribonucleic acid (mRNA) vaccines, designed for decentralized, thermostable delivery. The MVP integrates automated dispensing, vacuum-driven mold loading, and drying, ensuring consistent MNA formation and high cargo efficiency. Using a combination of PVA and PVP, the system stabilizes LNPs and maintains vaccine efficacy for at least 6 months at room temperature. MNAs fabricated with mRNA encoding the SARS-CoV-2 spike protein receptor-binding domain induced immune responses in mice comparable to intramuscular injections. The patches demonstrated effective tissue penetration, mechanical integrity, and rapid dissolution, with a tip-loading process minimizing vaccine waste. A modeling approach confirmed that human vaccine doses could be delivered through small, scalable patches. This innovation enables self-administration, eliminates cold-chain requirements, and supports rapid, local vaccine production in low-resource settings. The printed patches were stable at room temperature for months [132]. Although this particular application is transdermal (skin patch for vaccination), the underlying technology—a portable 3D printer for MNAs—could be repurposed for non-transdermal needs as well. For example, in a remote clinic, one could print a bespoke MNA catheter for an emergency arterial intervention or print dissolvable MNA inserts for an inner ear drug delivery on site. The concept of decentralized manufacturing of MNAs underscores the flexibility introduced by 3D printing: it can shorten development cycles and allow tailoring of the MNA device to the patient or scenario [132].

In another study, Che et al. [133] developed innovative 4D-printed MNAs using a dual-sensitive biopolymer, hydroxybutyl methacrylated chitosan (HBCMA), via DLP 3D printing. The study synthesized HBCMA and optimized its photorheological and mechanical properties for use as a responsive ink. The resulting MNAs exhibited dynamic geometry changes—swelling or shrinking—based on temperature variations, enhancing needle sharpness and mechanical strength. Through layer thickness control and anti-aliasing, high-resolution fabrication was achieved. The needles demonstrated sufficient penetration into soft tissues (chicken breast) and supported the sustained release of rhodamine B, a model drug. Drug release kinetics followed the Korsmeyer–Peppas diffusion model. SEM confirmed the formation of temperature-dependent pore structures. This is the first report of 4D-printed MNAs using HBCMA, offering a promising approach for non-transdermal drug delivery applications targeting mucosal, vascular, or internal tissues.

In summary, the landscape of 3D-printed MNAs is rapidly expanding. From enabling first-in-kind treatments for hearing loss and sampling microbes in the gut to paving the way for on-site printing of personalized patches, these emerging directions illustrate the profound impact of 3D printing on MNA technology. They break traditional barriers of where and how MNAs can be used, heralding a future in which nearly any accessible organ or tissue could become a target for a smart MNA-based intervention.

As discussed earlier, the literature’s reliance on small-scale, brief studies is a recurring limitation. There are few examples of long-term or large-animal testing, and the majority of the evidence in favor of polymeric 3D-printed MNAs is still limited to in vitro, ex vivo, or small-animal experiments. Additionally, evaluations of biocompatibility usually focus on short-term responses rather than chronic degradation, immunological reactivity, or long-term mechanical fatigue. Without reporting and validation standards, extrapolating these results to human use remains speculative. These gaps must be addressed if clinical translation is to succeed and be approved by regulators.

Recent studies on non-transdermal uses of 3D printed polymeric MNAs are summarized in Table 2. Although the examples provided across organ systems show the exciting potential of these devices, it is important to note that, as this table shows, the majority of 3D-printed MNAs are still in the preclinical or proof-of-concept stage. Widespread clinical adoption will require overcoming challenges related to long-term safety, manufacturing viability, and regulatory approval. The section that follows discusses these translational difficulties in greater detail.

## 4. Challenges and Future Directions

While MNAs show significant potential to transform drug delivery, their widespread use depends on addressing key technical, regulatory, and manufacturing hurdles. Overcoming these issues will require joint efforts among researchers, industry, and regulatory authorities to create standardized procedures, affordable production methods, and thorough safety evaluations. Tackling these challenges will help MNAs move from promising innovations to routine clinical tools.

A major hurdle lies in navigating regulatory concerns. The approval process for 3D-printed MNAs is particularly intricate due to their dual role as both devices and drug delivery or diagnostic tools. This combined functionality demands adherence to various regulatory frameworks, making the path to approval more complex [62,134]. Securing regulatory approval typically involves proving that MNAs can reliably and safely penetrate to appropriate depths, avoiding contact with blood vessels or nerve endings. While research such as that by Defelippi et al. [135] has addressed depth regulation, there is still a clear need for more standardized guidelines.

The lack of well-defined regulatory standards for MNA technologies adds complexity to the approval process. This puts manufacturers in a difficult position, forcing them to choose between costly aseptic production and more affordable, low-bioburden methods that may raise regulatory concerns. Moreover, quality control practices are still evolving, with an urgent need for non-destructive, in-line testing techniques to maintain consistency. Until a unified regulatory framework is established, the approval and widespread commercialization of MNAs will likely remain delayed [136].

At present, MNA approvals are handled individually, without a consistent regulatory system, which results in extended licensing timelines and delays in bringing products to market. Addressing this requires the development of a comprehensive regulatory framework that accounts for critical factors such as geometry, formulation, sterilization, and packaging. Incorporating current good manufacturing practice (cGMP) standards and adopting a quality-by-design strategy could streamline approvals and enhance the overall reliability of MNAs as drug delivery solutions [137].

Safety concerns and strict regulatory requirements hinder the clinical translation of MNAs. To avoid negative reactions when MNAs enter the tissue, it is crucial to make sure the materials are biocompatible. It is still difficult to sterilize MNAs without sacrificing their structural integrity, especially for designs based on polymers [138]. It is important to carefully consider the long-term effects of using MNAs, including any possible tissue damage or infection risks from repeated use. Few studies have been conducted to date to address these issues, particularly for applications that need to be used repeatedly or for extended periods of time [139,140].

Clinical translation and trials present major obstacles to the future development of 3D-printed MNAs. It has proven difficult to implement these advancements in clinical practice, despite promising laboratory results. One of the biggest obstacles to regulatory approval and broad adoption is still the scarcity of long-term clinical data. Clinical studies have increased, and early-stage trials (Phase 1 and 2) have given way to later-stage trials (Phase 3 and 4). However, a significant number of trials are still unclassified, indicating regulatory uncertainty. There are not many large-scale trials to prove long-term safety and efficacy; most of the trials concentrate on drug delivery, mainly targeting tissue conditions, vaccines, and ocular treatments. Furthermore, there are fewer industry-sponsored trials than studies for academic research, which could postpone commercialization. To overcome these obstacles and expedite clinical adoption, academia, industry, and regulatory bodies must work together more closely [141].

MNAs also encounter notable difficulties in the realm of patents and intellectual property. A wide range of patents has already been granted for various aspects of 3D-printed MNAs [142,143,144]. The rising number of patents in this space underscores the highly competitive landscape of innovation. Excessively broad or restrictive patents can impede progress by restricting access and limiting competition. With patents now covering areas such as drug delivery, fabrication techniques, and device applications, navigating freedom to operate is becoming increasingly difficult. Furthermore, overlapping claims and broad cooperative patent classification (CPC) categories contribute to legal uncertainty, which may delay commercialization. Future progress will depend on clearer regulatory pathways and collaborative licensing approaches that support both innovation and accessibility, fostering long-term advancement in MNA technology [141]. Enhancing cooperation amongst patent offices, industry stakeholders, and regulatory bodies can also help define more precise rules for MNA patenting while striking a balance between innovation and market accessibility.

Scalability is another significant obstacle to the future development of 3D-printed MNAs. Even though 3D-printed MNAs have a lot of promise, increasing their production for broad clinical use is still a major obstacle. Large-scale manufacturing is hampered by the high production costs associated with 3D printing, which is dependent on specialized tools and materials. High-resolution 3D printers, for example, are frequently too costly and slow for use in commercial settings. Additionally, as production volumes rise, scalability depends on preserving constant quality. To ensure both safety and efficacy, it is essential to maintain consistency in material properties, tip sharpness, and dimensions across batches. The inherent complexity of production and the absence of standardized manufacturing techniques represent another major barrier. The manual, lab-scale fabrication techniques that many developers still use are unsuitable for large-scale manufacturing. The vast range of MNA formulations, designs, and applications calls for the creation of specialized machinery and creative production techniques [145]. To move MNAs from research prototypes to commercially viable medical devices that can be incorporated into larger healthcare markets, it is imperative that the scalability issues be resolved.

Scaling up the MNA production while keeping costs affordable is challenging. Compared to conventional manufacturing techniques like molding, 3D printing is more expensive per unit but offers greater customization and accuracy. This is particularly problematic for applications that need to be deployed on a large scale [43,61]. The total cost is increased by the need for specialized tools and knowledgeable workers to fabricate MNAs. In many areas, finding and keeping workers skilled in 3D printing technologies continues to be a challenge [134,146].

When introducing 3D-printed MNAs to clinical use, cost and manufacturing scalability continue to be important factors. Due to slower build rates, 3D printing typically has higher production costs per unit at scale when compared to conventional micromolding or lithographic techniques. However, by doing away with the need for molds or specific required tools, it provides remarkable cost savings during the prototyping and early development phases. In contrast to traditional methods, 3D printing allows for rapid iteration and design flexibility for non-transdermal applications that require complex geometries, variable drug loading, or integration with other devices like ingestible capsules or ocular inserts. Furthermore, point-of-care or decentralized manufacturing may be made easier by 3D printing, which could reduce supply chain and storage costs. Traditional methods may still be more cost-effective for large-scale manufacturing with standardized designs, but as automation and throughput in 3D printing technologies improve, the cost per unit is expected to decrease.

Also, the sensitivity of 3D printing processes to parameters like temperature, resin composition, and printing speed makes it difficult to maintain consistency across batches. Inconsistencies in needle dimensions and drug delivery performance may result from this variability [134,138]. The layer-by-layer process of 3D printing is slow by nature, which makes it less suitable for high-throughput production. Techniques like dual-molding, as they require more infrastructure, are faster alternatives, as investigated by Ren et al. [62,138].

MNA fabrication involves a number of fabrication challenges. It is difficult to achieve the desired geometry, particularly for the hollow needles’ inner and outer diameters (IDs and ODs). When it comes to creating incredibly fine and consistent bore sizes, even high-resolution 3D printing methods have limitations [134,139,146]. Even with the advances in 3D printing, there are still a number of obstacles to overcome when printing MNAs. Since even small flaws can impair fluid flow, consistent lumen formation in hollow MNAs continues to be a significant challenge. Another crucial consideration is material biocompatibility, particularly for long-term applications.

Developing high-throughput 3D printing methods without compromising resolution and consistency remains a challenge [145,147]. Advances in material purification and post-processing will be crucial for reducing cytotoxicity [78,80]. Addressing the high costs of advanced 3D printing technologies like 2PP will be key to broader adoption [60,148]. Leveraging stimuli-responsive polymers and nanocomposites will enhance the multifunctionality of MNAs [100,102].

Looking ahead, several trends are likely to shape the future of 3D-printed MNAs. First, clinical translation and safety validation will be paramount. Many of the examples discussed are in the preclinical or prototype stages. Ensuring biocompatibility of new printable materials, sterility of manufacturing, and consistent performance in vivo will be necessary steps toward regulatory approval and clinical adoption. The encouraging results thus far (e.g., MNA vaccines and patches in animal models with minimal side effects [11]) suggest that with rigorous testing, these devices can meet safety standards.

Second, we anticipate a growth in combination devices—MNAs integrated with electronics or sensors. For diagnostic applications especially, an MNA could not only sample fluid but also immediately analyze it (for example, by coupling a 3D-printed hollow MNA with a miniaturized sensor in its lumen). Three-dimensional printing is already used to create intricate micro-electromechanical systems; combining that with MNAs could yield self-monitoring “smart” patches for organs. An example on the horizon is an MNA that delivers a drug and then measures the tissue’s response (e.g., inflammatory markers), all in one patch, enabling closed-loop therapy. Future research will likely focus on integrating artificial intelligence and wireless communication to create smart MNA systems capable of real-time decision-making and remote monitoring. MNAs are rapidly advancing beyond simple drug delivery devices, integrating with sensors and microfluidics to create multifunctional platforms for diagnostics, monitoring, and therapy. These hybrid systems are paving the way for personalized medicine, real-time health monitoring, and sustainable agriculture. This article explores the integration of MNAs with sensors and microfluidics, highlighting innovations and applications. Smart microfluidic systems use stimuli-responsive materials to control fluid dynamics and enhance drug delivery. For example, Dongre et al. [149] highlighted the evolution of MNAs integrated with biosensors and microfluidics, emphasizing their ability to provide continuous, controlled drug release based on patient needs. Microfluidic-enabled MNA platforms allow precise fluid management and analysis, expanding their use in point-of-care diagnostics and drug delivery.

Third, personalized medicine stands to benefit. As was previously mentioned, MNA devices can be tailored to fit particular anatomy or medical conditions through the use of 3D printing. Imagine a scenario where a patient’s magnetic resonance imaging (MRI) scan is used to design an MNA implant perfectly shaped to the resection cavity of a brain tumor—delivering chemotherapy precisely to where residual tumor cells remain. This level of customization could improve outcomes and reduce trial-and-error in dosing. Vander Straeten et al.’s MNA printer is a step in this direction, pointing to a future where pharmacies or hospitals could print bespoke MNA therapies on-site [131].

Finally, continued innovation in 4D printing and materials science will likely produce MNAs that are even smarter—for instance, needles that sense the local pH or temperature and adjust their drug release rate accordingly, or needles that intentionally biodegrade at a certain time after performing their function (to avoid any long-term tissue impact). The work by Che et al. [133] on dual-sensitive 4D-printed MNAs demonstrates this concept of needles that respond to stimuli (temperature, light) to improve their performance.

Stimuli-responsive materials, including smart polymers and nanocomposites, are unlocking new functionalities in MNAs. These materials can dynamically respond to environmental changes, such as pH or temperature, enabling targeted drug delivery and biosensing [100,101]. The integration of MNAs with microfluidics and sensors is revolutionizing their applications across healthcare, agriculture, and environmental monitoring. By enabling precise diagnostics and therapeutic interventions, these hybrid systems address critical challenges in modern medicine and technology. Continued innovation in materials and manufacturing will drive their adoption, making them indispensable tools for the future.

One main barrier to the clinical application of 3D-printed MNAs is still regulatory uncertainty. There are currently no device-specific regulations for MNAs, especially those made with 3D printing, in regulatory frameworks like those set up by the Food and Drug Administration (FDA) and European Medicines Agency (EMA). These include, but are not limited to, batch-to-batch reproducibility, quality control protocol for micro-scale property, and long-term biocompatibility tests of new polymers. Moreover, the variability brought about by the decentralized and adaptable nature of 3D printing in production environments makes standardization difficult. Regulatory authorities are becoming increasingly aware of the fact that existing guidelines such as ISO 10993 on biological assessment of medical devices [150] and ISO/ASTM 52900 on additive manufacturing fundamentals [151] need to be updated, such that the peculiar properties of 3D printing drug delivery systems can be considered. Manufacturing will remain a significant hurdle for clinical approval until more direct, efficient routes are developed.

In conclusion, 3D-printed MNAs represent a transformative convergence of bioengineering and medicine. By overcoming the historical limitations of drug delivery in hard-to-reach tissues, they open up new possibilities for treating diseases and monitoring health in ways that are more effective, less invasive, and more patient-friendly. As research continues to tackle the remaining challenges and refine these technologies, we can expect to see 3D-printed MNAs moving from laboratories into clinics—delivering vaccines without needles, treating chronic illnesses at the source, and enabling diagnostics that were once science fiction. The diverse applications reviewed herein—from the brain to the inner ear to the gut—underscore that the impact of 3D-printed MNAs will be felt across virtually all fields of medicine [14,124]. The next decade will be critical in translating these advances into real-world therapies and diagnostic tools, ultimately improving outcomes and quality of life for patients with conditions that are currently difficult to treat with conventional approaches.

## 5. Conclusions

Polymeric 3D-printed microneedle arrays (MNAs) are a revolutionary development in drug delivery and diagnostics that could be used for non-transdermal applications other than conventional transdermal administration. The fabrication of highly customized, high-resolution polymeric MNAs for a variety of anatomical sites has been made possible by recent advancements in 3D printing technologies, ranging from SLA and DLP to intricate processes like 2PP. Non-transdermal applications in the brain, mouth, eyes, gastrointestinal (GI) tract, cardiovascular system, and reproductive organs—areas where traditional delivery methods face major physiological challenges—are particularly impacted by these advancements.

There are still several challenges to be solved, despite the fact that early successes demonstrate that these systems are viable and promising. Some examples include improving the mechanical strength of biodegradable polymers, achieving precise and consistent drug release kinetics, ensuring safety in sensitive tissues, and meeting regulatory standards for clinical translation. Future research should focus on developing scalable manufacturing platforms to enable widespread clinical use, as well as integrating smart materials and biosensors into MNAs for real-time monitoring and feedback-controlled therapy.

In summary, polymeric 3D-printed MNAs offer patient-friendly, targeted, and minimally invasive solutions for the next generation of healthcare technologies for non-transdermal drug delivery and diagnostic applications.

## Figures and Tables

**Figure 1 polymers-17-01982-f001:**
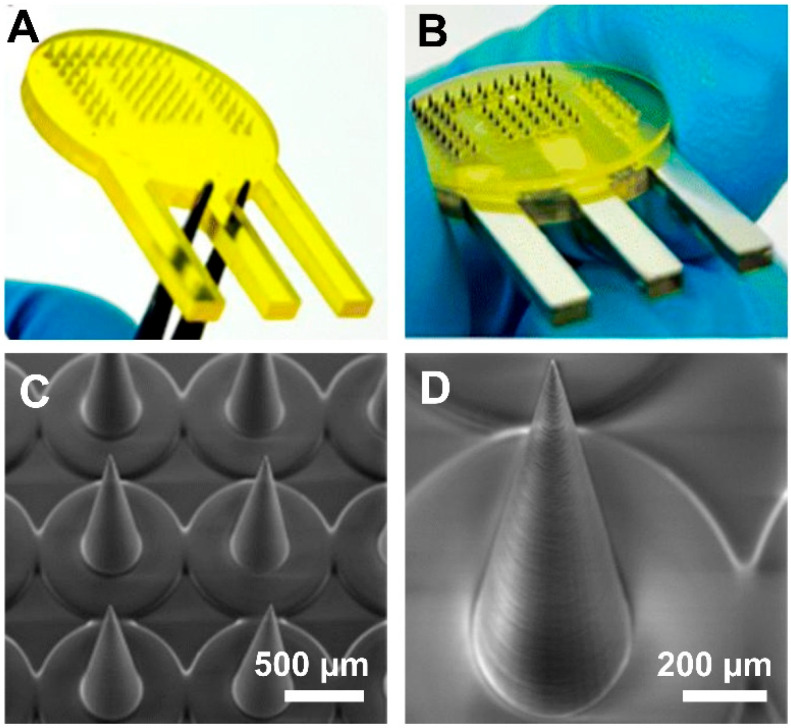
Fabrication and morphological characterization of PmSL SLA 3D-printed MNA. (**A**) As-printed MNA structure, (**B**) fully assembled device with inkjet-printed conductive paths for three-electrode electrochemical sensing, (**C**) scanning electron microscopy (SEM) image showing the details of the MNA, (**D**) high-magnification SEM image of a single MN illustrating its fine tip resolution, essential for effective tissue penetration. Reprinted under a Creative Commons Attribution 3.0 Unported license [43].

**Figure 2 polymers-17-01982-f002:**
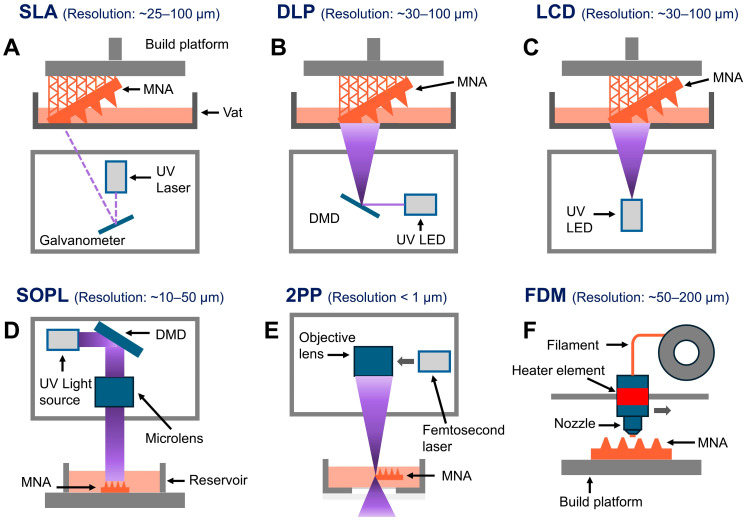
Schematic illustration of the working principles of key 3D printing techniques used for polymeric MNA fabrication: (**A**) SLA, (**B**) DLP, (**C**) LCD, (**D**) SOPL, (**E**) 2PP, (**F**) FDM.

**Figure 3 polymers-17-01982-f003:**
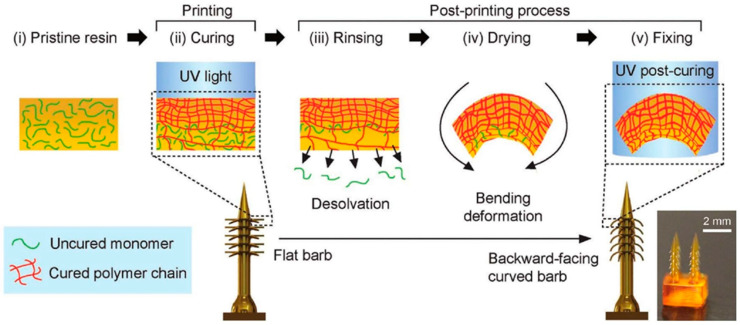
Schematic illustration of the 4D printing and post-processing workflow for barbed MNAs. (**i**) A pristine resin is selectively photopolymerized to form a crosslinked network, while uncured monomers remain in the structure. (**ii**) Rinsing removes the uncured monomers, leading to (**iii**) desolvation-driven shrinkage. (**iv**) Drying induces stress gradients across the cured regions, causing programmed bending deformation. (**v**) A final UV post-curing step fixes the structure into its backward-facing, curved barb configuration. Reproduced with permission from Wiley [95].

**Figure 4 polymers-17-01982-f004:**
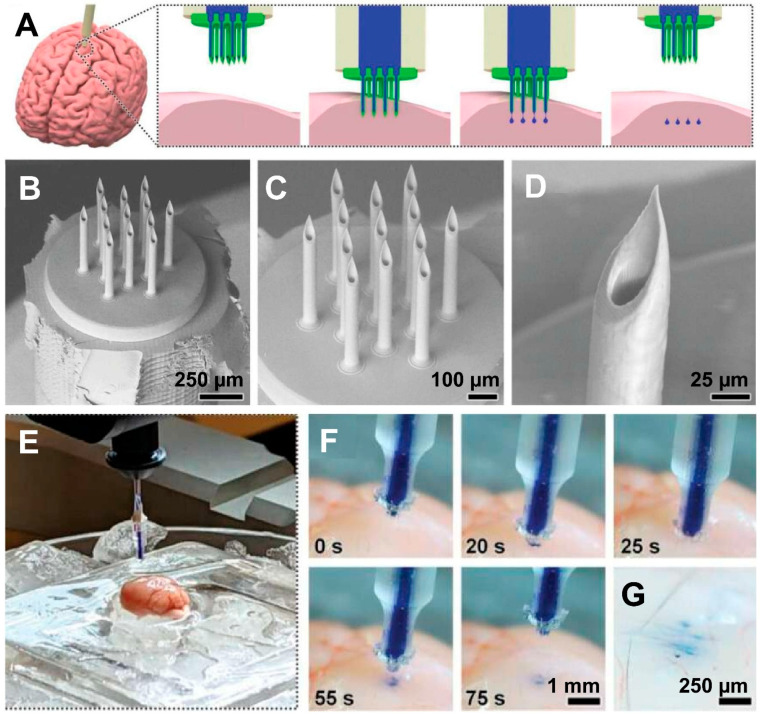
A developed MNA system for drug delivery to the brain. (**A**) Schematic illustration of the application of integrating MNA–capillary assemblies with nanoinjector systems to facilitate MNA-mediated simultaneous, distributed microinjections of target fluidic substances/suspensions into brain tissue. (**B**–**D**) SEM images of the fabricated MNA with different magnifications. (**E**) Experimental setup for the evaluation of the drug delivery capability of the developed MNAs, including the MNA–capillary assembly interfaced with a custom-built nanoinjector and an excised mouse brain on ice. (**F**) Sequential images of a representative MNA penetration, microinjection, and retraction process for a surrogate fluid (blue-dyed deionized water) injected into brain tissue. (**G**) Magnified view of the postinjection site. Adopted with permission from Wiley [17].

**Figure 5 polymers-17-01982-f005:**
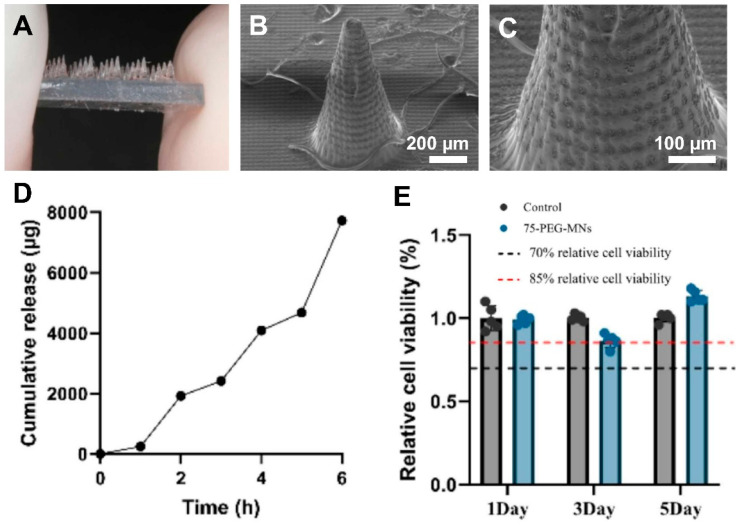
A developed MNA for drug delivery to the oral cavity. (**A**) Morphology of hydrogel MNA. (**B**,**C**) SEM images of the needle of the fabricated MNA with different magnifications. (**D**) Drug release profile showing that approximately 3.2% of madecassoside was released within 6 h, totaling 7.73 mg. (**E**) Effect of hydrogel MNA extracts on cell viability, showing a decrease of about 15% at day 3, which represented the lowest observed viability. Adopted under a CC-BY license [75].

**Figure 6 polymers-17-01982-f006:**
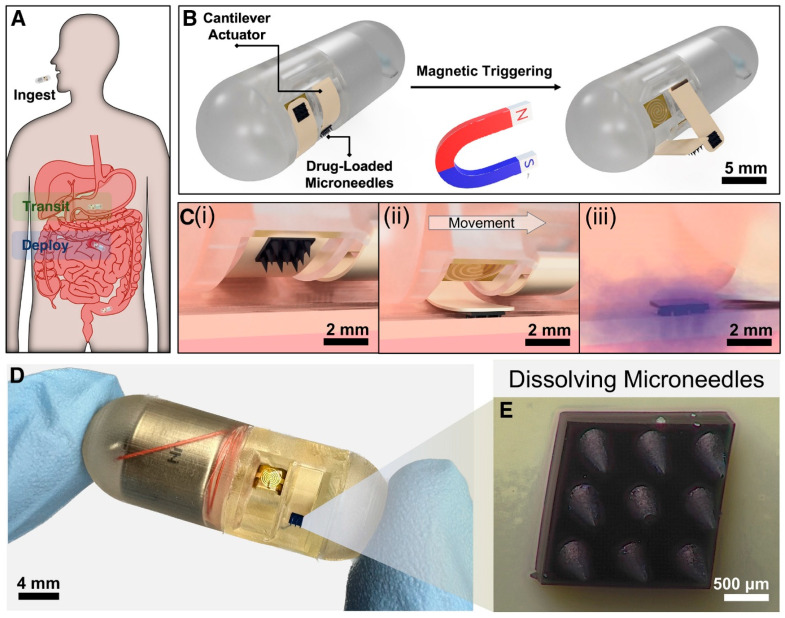
(**A**) Stages of capsule function, from ingestion to deployment and eventual excretion. (**B**) Schematic of the magnetic actuation mechanism, illustrating how the cantilever deploys in response to an external magnetic field. (**C**) Detailed view of cantilever (**i**) before deployment, (**ii**) during the deployment of drug-loaded MNAs into the tissue, and (**iii**) during the subsequent release of the drug. (**D**) Photograph of the fully assembled capsule. (**E**) Close-up of the dissolvable microneedle array designed to release the drug into GI tissue upon deployment. Reprinted with permission from Cell Press [119].

**Figure 7 polymers-17-01982-f007:**
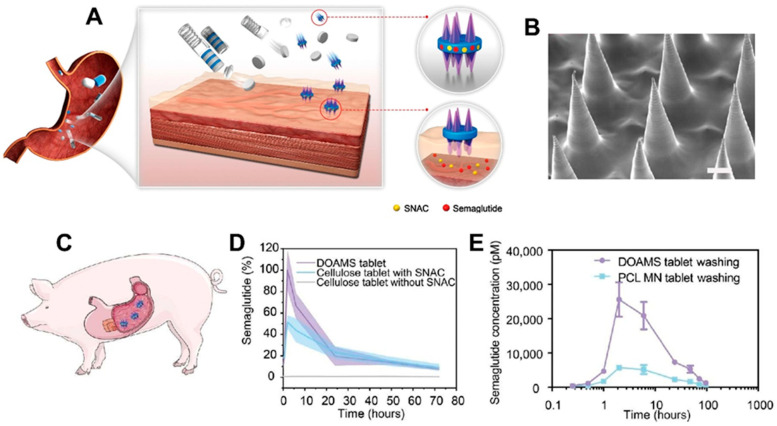
Schematic and experimental validation of the DOAMS MNA tablet system for gastric drug delivery. (**A**) Diagram showing pH-triggered deployment and anchoring of the tablet in the stomach. (**B**) SEM image of MNA. (**C**) Illustration of tablet delivery in pigs. (**D**,**E**) Plasma semaglutide profiles showing enhanced absorption with DOAMS compared to control tablets. Reprinted under a CC BY license [123].

**Table 1 polymers-17-01982-t001:** Advantages and limitations of popular 3D printing methods for MNA fabrication.

Fabrication Technique	Advantages	Limitations	References
SLA	High-resolution, smooth surface finishes, ideal for intricate designs.	Slower speed com pared to DLP, limited build volume.	[16,32,33,34,35,36,37,38,39,40,41,42,43,74]
DLP	Faster printing speed, high resolution, suitable for intricate designs.	Potential pixelation effects.	[9,18,42,47,48,49,50,75]
LCD	Affordable, large build volume, avoids pixel distortion.	Slightly lower resolution and accuracy compared to DLP.	[41,53,54,55]
SOPL	High precision, efficient for specific patterns, suitable for intricate designs.	Limited flexibility in pattern changes during printing.	[25]
2PP	Extremely high resolution, suitable for nanoscale features.	Expensive, slow printing speed, limited material options.	[22,59,60,61,62,63,64]
FDM	Cost-effective, wide range of materials, user-friendly.	Low resolution. Often requires post-fabrication processes	[68,69,70,71]

**Table 2 polymers-17-01982-t002:** Recent research conducted on non-transdermal applications of 3D-printed polymeric MNAs.

Organ/Tissue Applied	Results	Study Type	Model System	Development Stage	Reference Number
Brain	Reduced tumor size and increased survival via spatiotemporal multidrug release using silk microneedle patch.	In vivo	Mice	Preclinical	[107]
	Enabled precise sequential release of multiple drugs, significantly inhibiting tumor growth and prolonging survival.	In vivo	Rodent	Preclinical	[112]
	Spatiotemporal multidrug release led to reduced tumor volume and significantly prolonged survival in GBM-bearing mice.	In vivo	Rodent	Preclinical	[113]
	Achieved controlled release and deep tissue penetration of nanoparticles using dissolving microneedles.	Ex vivo/In vitro	Rat brain tissue	Proof-of-concept	[114]
	Enabled precise and minimally invasive delivery of biomolecules into brain tissue.	Ex vivo, Simulation	Mouse brain tissue	Proof-of-concept	[17]
Oral cavity	Promoted gingival regeneration by enhancing fibroblast proliferation and collagen deposition using madecassoside-loaded MNAs.	In vitro, In vivo	Rabbits	Preclinical	[75]
Ocular (Eye)	Enabled precise intrascleral delivery using hollow MNAs and 3D-printed adapters with minimal tissue disruption.	Ex vivo	Porcine eye tissue	Proof-of-concept	[118]
	Achieved precise penetration with optimized MN geometry and SLA parameters for ocular patch fabrication.	Ex vivo	Porcine corneal and scleral tissues	Proof-of-concept	[37]
GI Tract	Enabled oral insulin delivery with significant systemic absorption using ingestible unfolding MNA system.	In vivo, Ex vivo	Human intestinal tissue (ex vivo); pigs (in vivo)	Preclinical	[120]
	Enabled rapid, targeted drug delivery with magnetically triggered MNA deployment and improved diffusion.	Ex vivo	Porcine intestinal tissue	Proof-of-concept	[119]
	Achieved anchored, sustained protein delivery with significant improvement in macromolecule absorption and glycemic control.	In vivo	Pigs	Preclinical	[123]
Inner ear	Enabled safe and precise HRWM perforation, with minimal force and structural integrity maintained.	Ex vivo	Human temporal bone tissue	Proof-of-concept	[129]
Point-of-care 3D printing	Induced strong immune response with thermostable mRNA-loaded MNAs and demonstrated vaccine dose scalability.	In vivo	Mice	Preclinical	[131]
	Demonstrated temperature-responsive shape change and sustained drug release from 4D-printed HBCMA MNAs.	Ex vivo	Chicken breast tissue	Proof-of-concept	[133]

## Data Availability

No new data were created or analyzed in this study. Data sharing is not applicable to this article.

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
