# Peer review of "Polymeric 3D-Printed Microneedle Arrays for Non-Transdermal Drug Delivery and Diagnostics"

_polymers, 2025, doi:10.3390/polym17141982_

Round 1

Reviewer 1 Report

Comments and Suggestions for Authors

Congratulations on your very thorough presentation of the topic.

Author Response

Reviewer comment: Congratulations on your very thorough presentation of the topic.

Response: Thank you for their positive feedback and pleased to hear that you found the presentation of the topic thorough and informative.

Reviewer 2 Report

Comments and Suggestions for Authors This is a highly valuable and comprehensive review highlighting the significant potential of polymeric 3D-printed MNAs for non-transdermal applications. Its major features lie in its extensive technical coverage of fabrication methods, materials, and application-specific design strategies, coupled with insightful identification of key challenges and future directions. The focus on a non-transdermal area is particularly commendable. The main weaknesses involve a relatively soft critique of the limitations of the primary studies referenced, the dense presentation of the application table, and the inherent difficulty in balancing the exciting potential with the current early-stage reality of many applications. Overall, it serves as an excellent resource for researchers and engineers advancing this rapidly evolving field. The author organizes the contents well following a logical sequence, which helps the authors to understand the points well. This review fills an important gap in non-transdermal microneedles for advanced drug delivery. The material limits, regulation hurdles, and production cost will determine its clinical adoption. Some aspects of the review should be considered before publication. (1) While challenges are listed, the critique of the limitations of the studies cited (e.g., small sample sizes, animal models vs. human applicability, long-term biocompatibility data gaps in the referenced works) are superficial. The review could more critically evaluate the evidence presented in the primary research it summarizes. (2) Table 2 summarizing applications is comprehensive but text-heavy and visually dense, making specific information retrieval potentially difficult. Consideration of presenting key results more succinctly could improve its utility. (3) Some points about the advantages of 3D printing for customization and overcoming barriers are reiterated multiple times throughout different sections, slightly affecting conciseness. (4) While highlighting the potential of 3D printed polymeric MNAs is an advantage, the review unavoidably reflects the field's current state: many applications discussed are still preclinical or early-stage prototypes. The emphasis on potential might overshadow the significant remaining hurdles for widespread clinical adoption. A clearer distinction between proven feasibility and aspirational future applications could be strengthened. The challenges section acknowledges this, but the aspirational tone dominates the applications section. (5) New kinds of materials, such as stimuli-responsive materials, including smart polymers, how they are combined with current AI-driven technologies may be considered. (6) The novel technologies for MNAs construction such as 2PP, LCD, SOPL, 4D-printing are introduced and compared in a Table. The working principles and processing may be expressed by providing schematic diagrams. (7) The images in this review seem less on such as the 3D-printed MNAs models, products, or fabrication techniques, as well as the application samples on the organs and therapy. Additional images or illustrations are encouraged to help the readers understand the content more clearly. (8) Regulatory concerns as a major hurdle for 3D-printed MNAs in future development may be explained referenced from national regulation, material safety standards or quality control variabilities. (9) A detailed cost evaluation comparing 3D-printed MNAs products to other conventional products or methods may be provided in terms of raw materials, production processes, and maintenance in clinical adoption. Although cost and scalability are mentioned as challenges, a more detailed analysis contrasting the cost-effectiveness and manufacturability of 3D printed MNAs versus traditional manufacturing for specific non-transdermal applications at scale is lacking.

Author Response

COMMENTS TO THE AUTHOR:

This is a highly valuable and comprehensive review highlighting the significant potential of polymeric 3D-printed MNAs for non-transdermal applications. Its major features lie in its extensive technical coverage of fabrication methods, materials, and application-specific design strategies, coupled with insightful identification of key challenges and future directions. The focus on a non-transdermal area is particularly commendable. The main weaknesses involve a relatively soft critique of the limitations of the primary studies referenced, the dense presentation of the application table, and the inherent difficulty in balancing the exciting potential with the current early-stage reality of many applications. Overall, it serves as an excellent resource for researchers and engineers advancing this rapidly evolving field. The author organizes the contents well following a logical sequence, which helps the authors to understand the points well. This review fills an important gap in non-transdermal microneedles for advanced drug delivery. The material limits, regulation hurdles, and production cost will determine its clinical adoption. Some aspects of the review should be considered before publication.

Response: Thank you for your thoughtful and encouraging evaluation of the manuscript. I greatly appreciate the recognition of the efforts to provide a comprehensive and well-structured review focused on non-transdermal applications of polymeric 3D-printed microneedle arrays (MNAs). Also, thank you for highlighting areas that could benefit from further clarification or elaboration. In response, revisions have been made based on your valuable comments.

(1) While challenges are listed, the critique of the limitations of the studies cited (e.g., small sample sizes, animal models vs. human applicability, long-term biocompatibility data gaps in the referenced works) are superficial. The review could more critically evaluate the evidence presented in the primary research it summarizes.

Response: Thank you for this insightful comment. In response, several parts of the manuscript to incorporate a more critical evaluation of the primary studies cited have been revised. Specifically, in sections addressing the brain/CNS (Section 3.1), gastrointestinal tract (Section 3.4), and reproductive system (Section 3.6), some descriptions and discussions have been added. Also, to put these issues in a broader context, a new paragraph was added at the end of Section 3.7.3.

[Pages 12-13]

Despite encouraging preclinical results, it should be highlighted that most of the studies discussed in this section for the brain/CNS drug delivery and diagnostic application are still in the proof-of-concept stage and mostly use animal models or simulated brain environments. These models fall short in capturing the complexity of human neuroanatomy, immunological responses, and tissue healing dynamics. Sample sizes are usually small, and little is known about the long-term biocompatibility, degradation, and potential neurotoxicity of implanted polymers. The absence of large-animal or early-phase clinical data significantly restricts our understanding of how these technologies will work in humans.

[Page 19]

The translational pathway remains unclear despite the promising performance of 3D-printed MNA capsules in early in vivo studies and ex vivo animal models. Several systems under discussion are still in the early phases of development and have not yet been proven to work on humans or in larger animal models. Additionally, issues like mechanical safety under peristaltic forces, mucosal immune responses to embedded polymers, and long-term tissue interactions are rarely addressed. As a result, caution should be used when applying recent research to clinical settings.

[Page 23]

It should be highlighted that the majority of studies are restricted to small animal models or in vitro experimental setups, despite the fact that research and studies have demonstrated encouraging results for local drug and vaccine delivery and detection using MNA systems in reproductive tissues. These results are preliminary, and little is known about the effects on human tissue, especially with respect to biodegradation, chronic exposure, and mucosal sensitivity. There are currently no established regulatory standards for microneedle systems designed specifically for reproduction, and there are few long-term safety and effectiveness data available.

[Page 25]

As discussed earlier, the literature's reliance on small-scale, brief studies is a recurring limitation. There are few examples of long-term or large-animal testing, and the majority of the evidence in favor of polymeric 3D-printed MNAs is still limited to in vitro, ex vivo, or small animal experiments. Additionally, evaluations of biocompatibility usually focus on short-term responses rather than chronic degradation, immunological reactivity, or long-term mechanical fatigue. Without reporting and validation standards, extrapolating these results to human use remains speculative. These gaps must be addressed if clinical translation is to succeed and be approved by regulators.

(2) Table 2 summarizing applications is comprehensive but text-heavy and visually dense, making specific information retrieval potentially difficult. Consideration of presenting key results more succinctly could improve its utility.

Response: Thank you for this valuable suggestion. In response, Table 2 has been revised to improve clarity and ease of use. Specifically, the table was structured to group applications by organ system for clearer navigation. Also, the amount of descriptive text in each cell was reduced. Furthermore, columns were added for study type (e.g., in vitro, in vivo, ex vivo, simulation) and model system (e.g., mouse, porcine tissue), and development stage (e.g., proof-of-concept, preclinical, translational)  to help readers quickly assess relevance.

[Page 25-26]

Table 2. recent research conducted on non-transdermal applications of 3D printed polymeric MNAs.

Organ/ Tissue applied

Results

Study Type

Model System

Development Stage

Reference number

Brain

Reduced tumor size and increased survival via spatiotemporal multidrug release using silk microneedle patch.

In vivo

Mice

Preclinical

[107]

Enabled precise sequential release of multiple drugs, significantly inhibiting tumor growth and prolonging survival.

In vivo

Rodent

Preclinical

[112]

Spatiotemporal multidrug release led to reduced tumor volume and significantly prolonged survival in GBM-bearing mice.

In vivo

Rodent

Preclinical

[113]

Achieved controlled release and deep tissue penetration of nanoparticles using dissolving microneedles.

Ex vivo/ In vitro

Rat brain tissue

Proof-of-concept

[114]

Enabled precise and minimally invasive delivery of biomolecules into brain tissue

Ex vivo, Simulation

Mouse brain tissue

Proof-of-concept

[17]

Oral cavity

Promoted gingival regeneration by enhancing fibroblast proliferation and collagen deposition using madecassoside-loaded MNAs.

In vitro, In vivo

Rabbits

Preclinical

[75]

Ocular (Eye)

Enabled precise intrascleral delivery using hollow MNAs and 3D-printed adapters with minimal tissue disruption.

Ex vivo

Porcine eye tissue

Proof-of-concept

[118]

Achieved precise penetration with optimized MN geometry and SLA parameters for ocular patch fabrication..

Ex vivo

Porcine corneal and scleral tissues

Proof-of-concept

[37]

GI Tract

Enabled oral insulin delivery with significant systemic absorption using ingestible unfolding MNA system.

In vivo, Ex vivo

Human intestinal tissue (ex vivo); pigs (in vivo)

Preclinical

[120]

Enabled rapid, targeted drug delivery with magnetically triggered MNA deployment and improved diffusion.

Ex vivo

Porcine intestinal tissue

Proof-of-concept

[119]

Achieved anchored, sustained protein delivery with significant improvement in macromolecule absorption and glycemic control..

In vivo

Pigs

Preclinical

[123]

Inner ear

Enabled safe and precise HRWM perforation with minimal force and structural integrity maintained

Ex vivo

Human temporal bone tissue

Proof-of-concept

[129]

Point-of-care 3D printing

Induced strong immune response with thermostable mRNA-loaded MNAs and demonstrated vaccine dose scalability.

In vivo

Mice

Preclinical

[131]

Demonstrated temperature-responsive shape change and sustained drug release from 4D-printed HBCMA MNAs.

Ex vivo

Chicken breast tissue

Proof-of-concept

[133]

(3) Some points about the advantages of 3D printing for customization and overcoming barriers are reiterated multiple times throughout different sections, slightly affecting conciseness.

Response: Thank you for this helpful observation. The manuscript is reviewed and identified instances where the benefits of 3D printing—particularly regarding customization, and overcoming barriers—were redundantly discussed across multiple application sections. To improve conciseness and readability repetitive statements are condensed and removed some redundant phrasing.

[Page 16]

The aforementioned advantages of 3D printing facilitate the development of polymeric MNAs with various geometries and drug-loading profiles that are appropriate for specific ocular conditions such as keratitis, glaucoma, and diabetic retinopathy.

[Page 17]

Capsules can be shaped using 3D printing to increase the likelihood that they will correctly align and make contact with the epithelium (for example, a self-righting shape that consistently lands needle-side down on the gut wall)

[Page 29]

As was previously mentioned, MNA devices can be tailored to fit particular anatomy or medical conditions through the use of 3D printing.

(4) While highlighting the potential of 3D printed polymeric MNAs is an advantage, the review unavoidably reflects the field's current state: many applications discussed are still preclinical or early-stage prototypes. The emphasis on potential might overshadow the significant remaining hurdles for widespread clinical adoption. A clearer distinction between proven feasibility and aspirational future applications could be strengthened. The challenges section acknowledges this, but the aspirational tone dominates the applications section.

Response: Thank you for your thoughtful and constructive comment. To address this point, the development stage have been added (e.g., preclinical, proof-of-concept) for each study in the revised Table 2, helping readers distinguish between validated and early-stage work. Also, a transitional paragraph has been added at the end of Section 3.7.3 to signal the contrast between promise and reality and to segue into the more critical discussion in the challenges section..

[Page 25]

Recent studies on non-transdermal uses of 3D printed polymeric MNAs are summarized in Table 2. Although the examples provided across organ systems show the exciting potential of these devices, it is important to note that, as this Table shows, the majority of 3D-printed MNAs are still in the preclinical or proof-of-concept stage. Widespread clinical adoption will require overcoming challenges related to long-term safety, manufacturing viability, and regulatory approval. The section that follows goes into greater detail about these translational difficulties.

(5) New kinds of materials, such as stimuli-responsive materials, including smart polymers, how they are combined with current AI-driven technologies may be considered.

Response: Thank the reviewer for this insightful suggestion. In response, a new paragraph discussing the integration of smart materials with AI-driven technologies, particularly in the context of real-time biosensing, data-informed actuation, and closed-loop therapeutic systems is added.

[Pages 8-9]

An exciting new area in intelligent drug delivery is the combination of AI-driven systems and stimuli-responsive materials. For instance, closed-loop insulin delivery can be supported by combining AI-enabled glucose monitoring systems with glucose-sensitive microneedles made of pH- or enzyme-responsive polymers. MNAs can adjust in real time to each patient's physiological state by using machine learning algorithms to evaluate biosensor data and optimize actuation timing or dosage. These kinds of combinations might serve as the foundation for therapeutic platforms that are genuinely autonomous.

(6) The novel technologies for MNAs construction such as 2PP, LCD, SOPL, 4D-printing are introduced and compared in a Table. The working principles and processing may be expressed by providing schematic diagrams.

Response: Thank you for this valuable recommendation. To enhance clarity and reader engagement, a schematic illustrations showing the working principles and processing steps of key advanced fabrication techniques has been added, including SLA, DLP, LCD, SOPL, 2PP, and FDM. Each subpanel highlights the core components of the process (e.g., light sources, motion systems, build orientation), the direction of printing, and the expected resolution range. To address the reviewer's comment regarding 4D printing, a figure is adopted and included a representative schematic (now presented as Figure 3) based on a previously published study (Zhang et al., Adv. Funct. Mater., 2020).

[Page 5]

Also, Figure 2 shows the fundamentals of the main 3D printing processes used in the fabrication of polymeric MNAs. These include extrusion-based (FDM), projection-based (DLP, LCD, SOPL), and laser-based (SLA, 2PP) techniques. Variations in light sources and achievable resolutions are highlighted in the schematic

[Page 9]

Figure 3 illustrastes the 4D printing and post-processing workflow for their barbed MNAs Schematically.

(7) The images in this review seem less on such as the 3D-printed MNAs models, products, or fabrication techniques, as well as the application samples on the organs and therapy. Additional images or illustrations are encouraged to help the readers understand the content more clearly.

Response: Thank you for your helpful comment regarding the need for more visual support in the manuscript. In response, two figure have been added to the manuscript:

  • A schematic comparison of the fundamentals of the main 3D printing processes used for MNA fabrication, such as SLA, DLP, LCD, SOPL, FDM, and 2PP, is shown in Figure 2.
  • Reproduced from Zhang et al. (Adv. Funct. Mater., 2020), Figure 3 depicts a typical 4D printing procedure for barbed microneedle arrays, including the steps of photopolymerization, desolvation, shape transformation, and fixation.

(8) Regulatory concerns as a major hurdle for 3D-printed MNAs in future development may be explained referenced from national regulation, material safety standards or quality control variabilities.

Response: Thank you for raising this important point. In response, a paragraph has beed added to the Challenges and Future Directions section to provide a discussion of regulatory hurdles affecting the clinical translation of 3D-printed polymeric MNAs.

[Page 29]

One main barrier to the clinical application of 3D-printed MNAs is still regulatory uncertainty. There are currently no device-specific regulations for MNAs, especially those made with 3D printing, in regulatory frameworks like those set up by the Food and Drug Administration (FDA) and European Medicines Agency (EMA). These include, but not limited to, batch-to-batch reproducibility, quality control protocol for micro-scale property, and long-term biocompatibility tests of new polymers. Moreover, the variability brought about by the decentralized and adaptable nature of 3D printing in production environments makes standardization difficult. Regulatory authorities are becoming increasingly aware of the fact that existing guidelines such as ISO 10993 on biological assessment of medical devices [150] and ISO/ASTM 52900 on additive manufacturing fundamentals [151] need to be updated, such that the peculiar properties of 3D printing drug delivery systems can be considered. Manufacturing will remain a significant hurdle for clinical approval until more direct, efficient routes are produced.

(9) A detailed cost evaluation comparing 3D-printed MNAs products to other conventional products or methods may be provided in terms of raw materials, production processes, and maintenance in clinical adoption. Although cost and scalability are mentioned as challenges, a more detailed analysis contrasting the cost-effectiveness and manufacturability of 3D printed MNAs versus traditional manufacturing for specific non-transdermal applications at scale is lacking

Response: Thank you for this excellent suggestion. To address this comment, a paragraph is added to the the Challenges and Future Directions section to include a a comparative analysis of the cost-effectiveness and manufacturability of 3D-printed MNAs versus conventional micromolding and lithography-based techniques.

[Pages 27-28]

When introducing 3D-printed MNAs to clinical use, cost and manufacturing scalability continue to be important factors. Due to slower build rates, 3D printing typically has higher production costs per unit at scale when compared to conventional micromolding or lithographic techniques. However, by doing away with the need for molds or specific required tools, it provides remarkable cost savings during the prototyping and early development phases. In contrast to traditional methods, 3D printing allows for rapid iteration and design flexibility for non-transdermal applications that require complex geometries, variable drug loading, or integration with other devices like ingestible capsules or ocular inserts. Furthermore, point-of-care or decentralized manufacturing may be made easier by 3D printing, which could reduce supply chain and storage costs. Traditional methods may still be more cost-effective for large-scale manufacturing with standardized designs, but as automation and throughput in 3D printing technologies improve, the cost per unit is expected to decrease.

Reviewer 3 Report

Comments and Suggestions for Authors

I carefully examined the manuscript polymers-3744523. In my opinion, this review has merit and can be published after some minor points have been addressed. The manuscript is somewhat similar to 10.1016/j.jconrel.2022.08.022 (ref 19 in the manuscript), but the focus on non-transdermal drug delivery grants sufficient novelty. Some aspects that require improvement are:

1) Section 2.1: I suggest placing here an explicative image showing the working principle of the 3D printing techniques explained.

2) Table 2: this table should report the minimum feature size achievable with each technique.

3) Section 2.2.5: in my opinion, the title “innovative materials” is too generic. Also composite resins or biodegradable materials can be innovative. I suggest splitting this section, for example, into “stimuli responsive materials” and “4D printing”.

4) Table 2: this table is somewhat useless. I suggest removing it.

Comments on the Quality of English Language

Even though I’m not fully qualified to evaluate this aspect, I think that the English of the manuscript requires some improvement.

Author Response

Comment: I carefully examined the manuscript polymers-3744523. In my opinion, this review has merit and can be published after some minor points have been addressed. The manuscript is somewhat similar to 10.1016/j.jconrel.2022.08.022 (ref 19 in the manuscript), but the focus on non-transdermal drug delivery grants sufficient novelty.

Response: Thank you for your careful evaluation and supportive assessment of our manuscript. Also, thank you for the acknowledgment of the manuscript’s novelty through its specific focus on non-transdermal applications of polymeric 3D-printed MNAs.

Some aspects that require improvement are:
1) Section 2.1: I suggest placing here an explicative image showing the working principle of the 3D printing techniques explained.

Response: Thank you for this helpful suggestion. To address this issue, a schematic illustrations showing the working principles and processing steps of key advanced fabrication techniques has been added, including SLA, DLP, LCD, SOPL, 2PP, and FDM. Each subpanel highlights the core components of the process (e.g., light sources, motion systems, build orientation), the direction of printing, and the expected resolution range.

[Page 5]

Also, Figure 2 shows the fundamentals of the main 3D printing processes used in the fabrication of polymeric MNAs. These include extrusion-based (FDM), projection-based (DLP, LCD, SOPL), and laser-based (SLA, 2PP) techniques. Variations in light sources and achievable resolutions are highlighted in the schematic.

[Page 6]

Figure 2. Schematic illustration of the working principles of key 3D printing techniques used for polymeric MNA fabrication (A) SLA, (B) DLP, (C) LCD, (D) SOPL, (E) 2PP, (F) FDM.

2) Table 2: this table should report the minimum feature size achievable with each technique.

Response: Thank you for this useful suggestion. In response, The resolution information has been added to Figure 2 to improve clarity and completeness. The figure includes relevant details regarding printing resolution and magnification, ensuring better understanding of the fabrication process and structural features of the developed MNA system.

3) Section 2.2.5: in my opinion, the title “innovative materials” is too generic. Also composite resins or biodegradable materials can be innovative. I suggest splitting this section, for example, into “stimuli responsive materials” and “4D printing”.

Response: Thank you for this excellent suggestion. To improve clarity and precision, the Section 2.2.5 is revised by splitting it into two subsections: “Stimuli-Responsive Materials” and “Materials for 4D Printed MNAs”.

2.2.5. Stimuli responsive materials

New materials that may be helpful in 3D-printed MNAs are being steadily discovered through ongoing research. These innovative materials are made to perform better, be more biocompatible, and have new features. These include smart polymers, which are substances that can change how they behave in response to environmental changes such as variations in temperature, pH, or light exposure [100]. Conductive polymers [101] can be utilized for fabricating MNAs that not only deliver drugs but also have sensing capabilities in applications like biosensing. By incorporating nanoparticles into polymer matrices, nanocomposites can significantly improve the antibacterial qualities of MNAs, such as silver nanoparticles, which help to prevent infections [102]. Novel materials have the potential to increase MNA technology's capabilities, opening up new applications and enhancing current ones.

An exciting new area in intelligent drug delivery is the combination of AI-driven systems and stimuli-responsive materials. For instance, closed-loop insulin delivery can be supported by combining AI-enabled glucose monitoring systems with glucose-sensitive microneedles made of pH- or enzyme-responsive polymers. MNAs can adjust in real time to each patient's physiological state by using machine learning algorithms to evaluate biosensor data and optimize actuation timing or dosage. These kinds of combinations might serve as the foundation for therapeutic platforms that are genuinely autonomous.

2.2.6. Materials for 4D Printed MNAs

The development of 4D printing in MNA fabrication requires the use of specialized materials. Unlike traditional 3D printing, 4D printing creates structures that can progressively change their shape, behavior, or properties in response to environmental stimuli like heat, moisture, or light. These changes occur because the materials are made to react and change in a specific, controlled way after printing. Time is the "fourth dimension" in this context because the printed object changes after it is first formed, allowing for dynamic, adaptive applications. In a study, a bioinspired MNA with backward-facing barbs for improved tissue adhesion was developed by Han et al. [95] using 4D printing. This technology made it possible to precisely create MNAs with intricate geometries, such as curved barbs, which are difficult to create with conventional techniques. These MNAs were created using projection microstereolithography (PµSL), which has programmable shape transformations that enable the barbs to curve backward when desolvated, greatly enhancing tissue adhesion. Figure 3 illustrastes the 4D printing and post-processing workflow for their barbed MNAs Schematically. The functionality of the MNAs was significantly influenced by the substance used in this procedure, particularly the photocurable resin (PEGDA 250). The barbs were able to bend and take on the required shape by varying the crosslinking density throughout the printing process. The creation of a highly effective MNA that exhibited 18 times stronger tissue adhesion than traditional MNAs was made possible by this carefully regulated material composition. Additionally, sustained delivery was demonstrated by drug release tests, underscoring the possibility of long-term uses in biosensing and drug delivery.

The type of material selected has a significant impact on the effectiveness, safety, and suitability of 3D-printed MNAs. Every alternative, including composites, biocompatible materials, photopolymer resins, and next-generation materials, has advantages and disadvantages of its own. A comprehensive understanding of these materials aids in the design of MNAs that meet the needs of particular medical applications. We can expect to see increasingly complex MNAs that expand their impact and utility in a range of clinical settings as 3D printing and material science develop further.

4) Table 2: this table is somewhat useless. I suggest removing it.

Response: Thank you for your feedback regarding Table 2. In response to a similar concern raised by Reviewer 2, the Table 2 is substantially revised to improve its clarity and utility. It would be appreciated if you could review the updated version and advise whether it still needs to be removed.

[Page 25-26]

Table 2. recent research conducted on non-transdermal applications of 3D printed polymeric MNAs.

Organ/ Tissue applied

Results

Study Type

Model System

Development Stage

Reference number

Brain

Reduced tumor size and increased survival via spatiotemporal multidrug release using silk microneedle patch.

In vivo

Mice

Preclinical

[107]

Enabled precise sequential release of multiple drugs, significantly inhibiting tumor growth and prolonging survival.

In vivo

Rodent

Preclinical

[112]

Spatiotemporal multidrug release led to reduced tumor volume and significantly prolonged survival in GBM-bearing mice.

In vivo

Rodent

Preclinical

[113]

Achieved controlled release and deep tissue penetration of nanoparticles using dissolving microneedles.

Ex vivo/ In vitro

Rat brain tissue

Proof-of-concept

[114]

Enabled precise and minimally invasive delivery of biomolecules into brain tissue

Ex vivo, Simulation

Mouse brain tissue

Proof-of-concept

[17]

Oral cavity

Promoted gingival regeneration by enhancing fibroblast proliferation and collagen deposition using madecassoside-loaded MNAs.

In vitro, In vivo

Rabbits

Preclinical

[75]

Ocular (Eye)

Enabled precise intrascleral delivery using hollow MNAs and 3D-printed adapters with minimal tissue disruption.

Ex vivo

Porcine eye tissue

Proof-of-concept

[118]

Achieved precise penetration with optimized MN geometry and SLA parameters for ocular patch fabrication..

Ex vivo

Porcine corneal and scleral tissues

Proof-of-concept

[37]

GI Tract

Enabled oral insulin delivery with significant systemic absorption using ingestible unfolding MNA system.

In vivo, Ex vivo

Human intestinal tissue (ex vivo); pigs (in vivo)

Preclinical

[120]

Enabled rapid, targeted drug delivery with magnetically triggered MNA deployment and improved diffusion.

Ex vivo

Porcine intestinal tissue

Proof-of-concept

[119]

Achieved anchored, sustained protein delivery with significant improvement in macromolecule absorption and glycemic control..

In vivo

Pigs

Preclinical

[123]

Inner ear

Enabled safe and precise HRWM perforation with minimal force and structural integrity maintained

Ex vivo

Human temporal bone tissue

Proof-of-concept

[129]

Point-of-care 3D printing

Induced strong immune response with thermostable mRNA-loaded MNAs and demonstrated vaccine dose scalability.

In vivo

Mice

Preclinical

[131]

Demonstrated temperature-responsive shape change and sustained drug release from 4D-printed HBCMA MNAs.

Ex vivo

Chicken breast tissue

Proof-of-concept

[133]